# Chiral gold nanoparticles enantioselectively rescue memory deficits in a mouse model of Alzheimer's disease

Ke Hou[1,2,3], Jing Zhao[4], Hui Wang[1], Bin Li[1,5], Kexin Li[6], Xinghua Shi [1], Kaiwei Wan [1], Jing Ai[6], Jiawei Lv[1], Dawei Wang[1], Qunxing Huang[1], Huayi Wang[1], Qin Cao[7], Shaoqin Liu[4✉] & Zhiyong Tang [1,3✉]

Preventing aggregation of amyloid beta (Aβ) peptides is a promising strategy for the treatment of Alzheimer's disease (AD), and gold nanoparticles have previously been explored as a potential anti-Aβ therapeutics. Here we design and prepare 3.3 nm L- and D-glutathione stabilized gold nanoparticles (denoted as L3.3 and D3.3, respectively). Both chiral nanoparticles are able to inhibit aggregation of Aβ42 and cross the blood-brain barrier (BBB) following intravenous administration without noticeable toxicity. D3.3 possesses a larger binding affinity to Aβ42 and higher brain biodistribution compared with its enantiomer L3.3, giving rise to stronger inhibition of Aβ42 fibrillation and better rescue of behavioral impairments in AD model mice. This conjugation of a small nanoparticle with chiral recognition moiety provides a potential therapeutic approach for AD.

[1] CAS Key Laboratory of Nanosystem and Hierarchical Fabrication, CAS Center for Excellence in Nanoscience, National Center for Nanoscience and Technology, Beijing 100190, China. [2] Center for Nanochemistry, Peking University, Beijing 100871, China. [3] University of Chinese Academy of Sciences, Beijing 100049, China. [4] MOE Key Laboratory of Micro-systems and Micro-structures Manufacturing, School of Life Science and Technology, Harbin Institute of Technology, Harbin 150001, China. [5] School of Chemical Engineering and Technology, Sun Yat-sen University, Zhuhai 519082, China. [6] Department of Pharmacology, the State-Province Key Laboratories of Biomedicine-Pharmaceutics of China, College of Pharmacy of Harbin Medical University, Harbin 150086, China. [7] Department of Chemistry and Biochemistry and Biological Chemistry, UCLA-DOE Institute and Howard Hughes Medical Institute, UCLA, Los Angeles, CA, USA. ✉email: shaoqinliu@hit.edu.cn; zytang@nanoctr.cn

Alzheimer's disease (AD), clinically characterized by progressive attrition of memory and other cognitive ability, is the most prevalent form of neurodegenerative diseases that affects 50 million people and costs US$ 1 trillion in 2018[1]. Generally, AD pathology involves proteinopathy of intra- or extracellular misfolded protein/peptide aggregates[2], metal ion dyshomeostasis[3], oxidative stress caused by reactive oxygen species[4], and loss of cholinergic transmission[5]. Increasing evidence indicate that deposition of neurofibrillary tangles and senile plaques, respectively composed of misfolded aggregates of intracellular microtubule-associated tau proteins and extracellular amyloid-β (Aβ) peptides, is two prominent histopathological features of AD[6]. Accordingly, targeting production, aggregation, and clearance of Aβ or tau from the brain is a mainstream measure for preventing or curing AD[7]. A wide variety of small molecules[8] and peptides[9] capable of inhibiting Aβ oligomerization or fibrillogenesis have been examined, and some improvements have been observed[7]. Unfortunately, many compounds enter into clinical trials but with moderate-to-poor success rate[10]. These failures originate from bad inhibition efficiency, low blood–brain barrier (BBB) permeability, unfavorable biocompatibility, and adverse effects, etc[11].

Alternatively, recent studies have demonstrated that some nanoparticles (NPs) could accelerate or retard Aβ aggregation process and thus be expected as potential drugs against AD[12–15]. The binding affinity and anti-amyloid capacity of NPs are tuned by controlling their physiochemical properties, such as size[12], shape[13], surface charge[14], and hydrophobicity[15]. Conjugation of NPs with peptides- and small molecule-based inhibitors further enhances their inhibition efficiency toward Aβ[16,17]. Among various NPs, gold (Au) NPs are believed as practical nanochaperones to inhibit and redirect Aβ fibrillization[14], owing to their good biocompatibility, easy functionalization[18], and potential ability to cross BBB[19]. However, by far the inhibition efficiency of the reported Au NP systems is much worse compared with small molecule- and peptide-based inhibitors[14]. Although grafting the peptides onto Au NPs might improve the inhibition activity against Aβ aggregations[20], their BBB permeability and in vivo efficacy remain unexplored.

In this work, chiral L- and D-glutathione (GSH) stabilized Au NPs with diameters of 3.3 nm, which are named as L3.3 and D3.3, are rationally designed. As a tripeptide antioxidant, GSH is known to protect tissues from damage caused by reactive oxygen species, and thus possesses the therapeutic potential in AD[21] and other neurodegenerative diseases[22]. In addition, due to the high level of GSH transporters located in the brain, surface capping with GSH enables Au NPs to permeate the BBB[23]. Most importantly, the surface chirality[24] and helix structure[25] can enantioselectively prevent Aβ aggregation in vitro. Therefore, the introduction of chiral GSH ligand is expected to endow gold NPs the exceptional ability toward chiral recognition of Aβ and enantioselective inhibition of Aβ fibrillation. Furthermore, Au NPs that function as the carriers will provide the unique opportunity for optimizing BBB permeability and in vivo efficacy via altering their size or shape.

## Results

**Inhibitory effect on Aβ42 aggregation in vitro.** Previous studies have demonstrated that the size of NPs exerts a large influence on Aβ aggregation[26] and such size-dependent phenomenon is likely associated with the curvature of NPs[27]. Thus, Monte Carlo simulation was first adopted to investigate the self-assembly behavior of peptide chains on different sized Au NPs[28] (see the details in Supplementary Method, Supplementary Fig. 1, and Supplementary Table 1). Interestingly enough, the presence of Au NPs would considerably decrease the average number of peptide chains in the formed aggregates, and the NPs with the size of 3–4 nm exhibit the best inhibition performance. So we intend to choose 3.3 nm Au NPs including L3.3 and D3.3 as the main investigated candidates[29]. As the contrast samples for inhibiting Aβ fibrillation, 9 nm and 15 nm L- or D-GSH-coated Au NPs (abbreviated as L9, D9, L15, and D15, respectively) are prepared for verifying the size effect[30] (Supplementary Figs. 2–4 and Supplementary Table 2). In addition, 3.5 nm non-chiral citrate-coated Au NPs (abbreviated as C3.5) are synthesized for investigating the enantioselective impact[31] (Supplementary Fig. 5). Aβ42 that has the largest AD toxicity among different Aβ peptides is selected for in vitro study[32].

The Thioflavin T (ThT) fluorescence assay was employed to inspect the inhibitory effect of different Au NPs on Aβ42 fibrillation. In the experiments, the concentration of various Au NPs is adjusted to have the same surface areas (Supplementary Table 3). When fresh Aβ42 alone is incubated at 37 °C (Supplementary Fig. 6), ThT fluorescence presents a sigmoidal shape as a function of incubation time (black curve in Fig. 1a), following the classic nucleation-growth process[15]. Remarkably, when incubating L3.3 or D3.3 with Aβ42 solution, a strong inhibition effect on Aβ42 fibrillization is discerned, as evidenced by a 60% or 63% decrease in the maximum ThT intensity compared with pure Aβ42 (magenta and blue curves in Fig. 1a). Also, 9 nm and 15 nm L- or D-GSH-coated Au NPs apparently inhibit Aβ42 fibrillization, in which the maximum ThT intensity decreases by 36% (L9), 58% (D9), 27% (L15), and 51% (D15), respectively (Supplementary Fig. 7a). To be more accurate, the lag time, the time required to reach half of the maximum fluorescence intensity $t_{1/2}$ and apparent aggregation constant $k$ are obtained by fitting ThT data with sigmoidal curves (Supplementary Table 4)[15]. It is clear that the chiral Au NPs exhibit an inhibition effect on Aβ42 aggregation by increasing the lag time and reducing the rate of Aβ42 aggregation. It deserves to be stressed that free L- or D-GSH molecules show negligible inhibition effect on Aβ42 fibrillization due to their weaker interaction with Aβ42 in solution[33] (Supplementary Fig. 8); while addition of C3.5 results in only 11% decrease in the maximum ThT intensity, indicating that C3.5 has very weak inhibitory effect like GSH molecules (green curve in Fig. 1a and Supplementary Table 4). Because the fluorescence quenching of ThT caused by Au NPs can be ignored under the experimental condition[12] (Supplementary Fig. 9), it is deduced that the introduction of chiral GSH stabilized Au NPs dramatically enhances the inhibition activity on Aβ42 fibrillization.

The influence of Au NPs on the conformational transition of Aβ42 was assessed using circular dichroism (CD) spectroscopy. As a control, Aβ42 aggregation leads to the formation of β sheet-rich framework with a typical positive CD peak at 195 nm and a negative minimum at 215 nm (black curve in Fig. 1b). All Au NPs might prevent the structural transition of Aβ42 from native random coil to β-sheet conformation in solution (Fig. 1b and Supplementary Fig. 7b), which is quantitatively estimated according to the previous report[34]. The Aβ42 fibrils contain $48.3 \pm 3.0\%$ of β-sheet structure, which reduces to $45.2 \pm 3.4\%$, $43.2 \pm 0.4\%$, $29.3 \pm 2.1\%$, $47.4 \pm 2.2\%$, $37.3 \pm 0.5\%$, $47.4 \pm 0.4\%$, and $42.2 \pm 2.0\%$ upon incubation with C3.5, L3.3, D3.3, L9, D9, L15, and D15, respectively. Correspondingly, the percentage of the random coil structures increases (Fig. 1c and Supplementary Fig. 7c).

Besides the conformational transition, the effect of Au NPs on the morphology change of Aβ42 aggregates is observed by atomic force microscopy (AFM) and transmission electron microscopy (TEM) imaging (Fig. 1d, e and Supplementary Fig. 7d, e). Typical non-branched amyloid fibrils with lengths of up to several

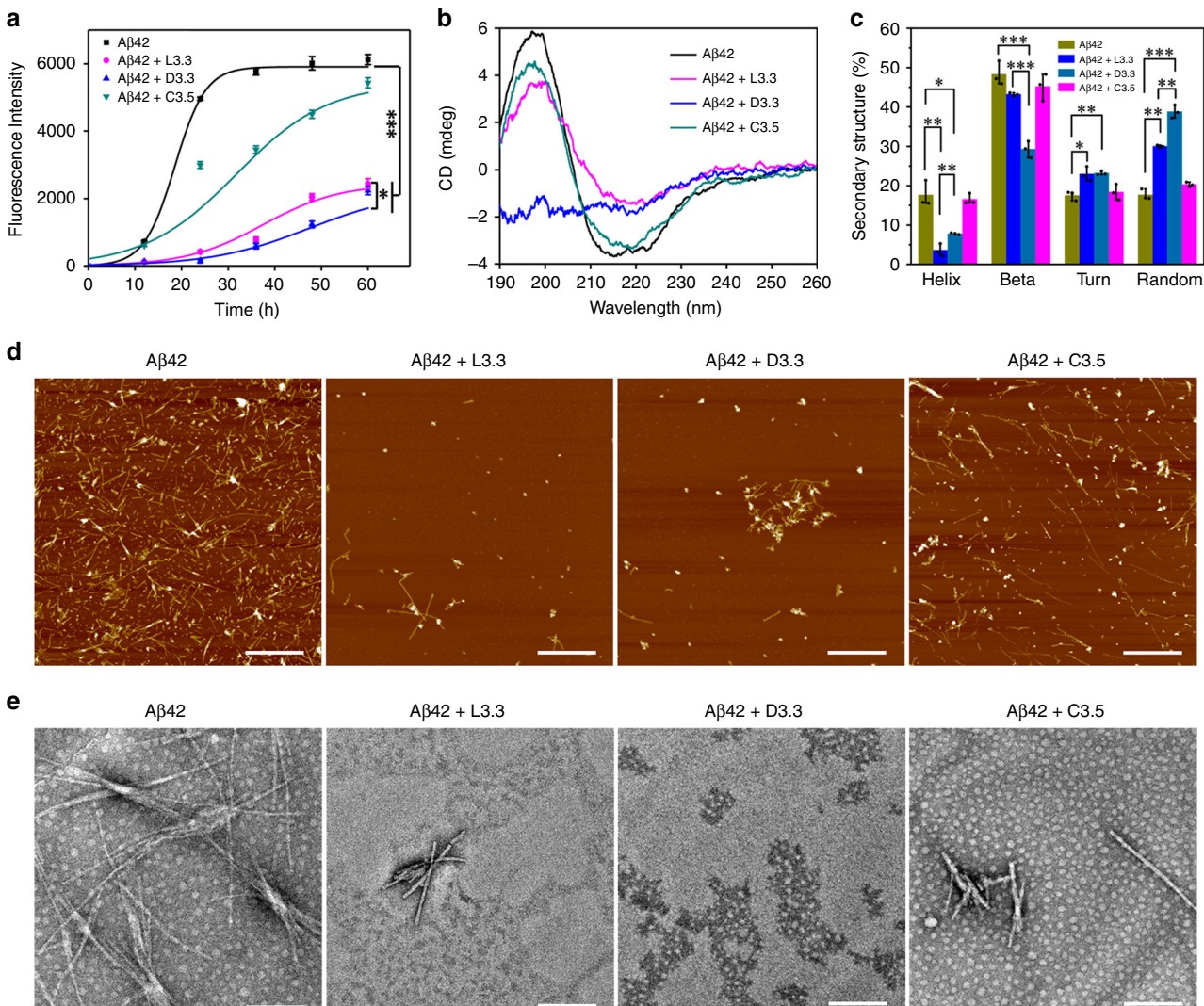

**Fig. 1 Effect of L3.3, D3.3, and C3.5 on Aβ42 fibrillization in vitro. a** ThT fluorescence assay of Aβ42 in the absence and presence of 3.3-nm L-GSH capped Au nanoparticles (denoted as Aβ42 + L3.3), 3.3 nm D-GSH capped Au nanoparticles (abbreviated as Aβ42 + D3.3), or 3.5 nm citrate-capped Au nanoparticles (denoted as Aβ42 + C3.5). The fibrillation kinetics is fitted with a sigmoidal function. **b** CD spectra of Aβ42 (40 μM) in the absence and presence of L3.3, D3.3, or C3.5 (110 nM) after co-incubation for 48 h. **c** Analysis of protein secondary structure. **d** AFM images of Aβ42 (40 μM) in the absence and presence of L3.3, D3.3, or C3.5 (110 nM) after co-incubation for 48 h. Scale bars, 1 μm. **e** TEM images of Aβ42 (40 μM) in the absence and presence of L3.3, D3.3, or C3.5 (110 nM) after co-incubation for 48 h. Scale bars, 200 nm. Error bars indicate the s.d. ($n = 3$ independent samples). *$P < 0.05$, **$P < 0.01$, ***$P < 0.001$, two-sided Student's $t$ test. For detailed statistical analysis see Supplementary Tables 6 and 7. Source data are provided as a Source Data file.

micrometers are found in the sample of pure Aβ42[35]. When incubating Aβ42 with various Au NPs, the length of the Aβ42 fibrils becomes shorter. Particularly, in the presence of D3.3, the mature amyloid fibrils are seldom discerned, and numerous small and relatively amorphous aggregates are distinguished, demonstrating their excellent efficacy on inhibiting Aβ42 fibrillization.

After validating the effectiveness of chiral Au NPs on inhibition of Aβ42 fibrillization, their size effect is studied by analyzing the CD spectra and ThT fluorescence assay (Supplementary Figs. 10 and 11). Regardless of L-GSH or D-GSH used as the surface stabilizers, 3.3 nm Au NPs display the best inhibition efficiency, confirmed by increasing the lag time and reducing the rate of Aβ42 aggregation[15]. This result is well consistent with the above simulation about the dependence of peptide aggregate structure on Au NP size (Supplementary Fig. 1 and Supplementary Table 1).

The surface chirality of GSH-coated Au NPs also strongly influences their inhibitory activity. D3.3 exerts a greater inhibitory effect on amyloid fibril formation than L3.3 (Fig. 1), and a similar phenomenon is recognized for 9 nm and 15 nm GSH-coated Au NPs (Supplementary Fig. 7). To reveal the origin of the difference in the performance of L3.3 and D3.3 on preventing Aβ42 aggregation, the binding constant between Aβ42 and L3.3 or D3.3 is measured with isothermal titration calorimetry (ITC)[36] (Fig. 2a, b). As displayed in Fig. 2c, the favorable enthalpy change ($\Delta H < 0$) and unfavorable entropy loss ($\Delta S < 0$) suggest that the complexation between Aβ42 and L3.3 or D3.3 is partly dominated by hydrogen bond and electrostatic interaction[37]. In addition, the fitted binding stoichiometry ($n_0$) implies that Aβ42 multilayers are formed on the surfaces of both L3.3 and D3.3[37]. The binding constant $K$ between Aβ42 and D3.3 is ~2.5 times larger than that between β42 and L3.3. Furthermore,

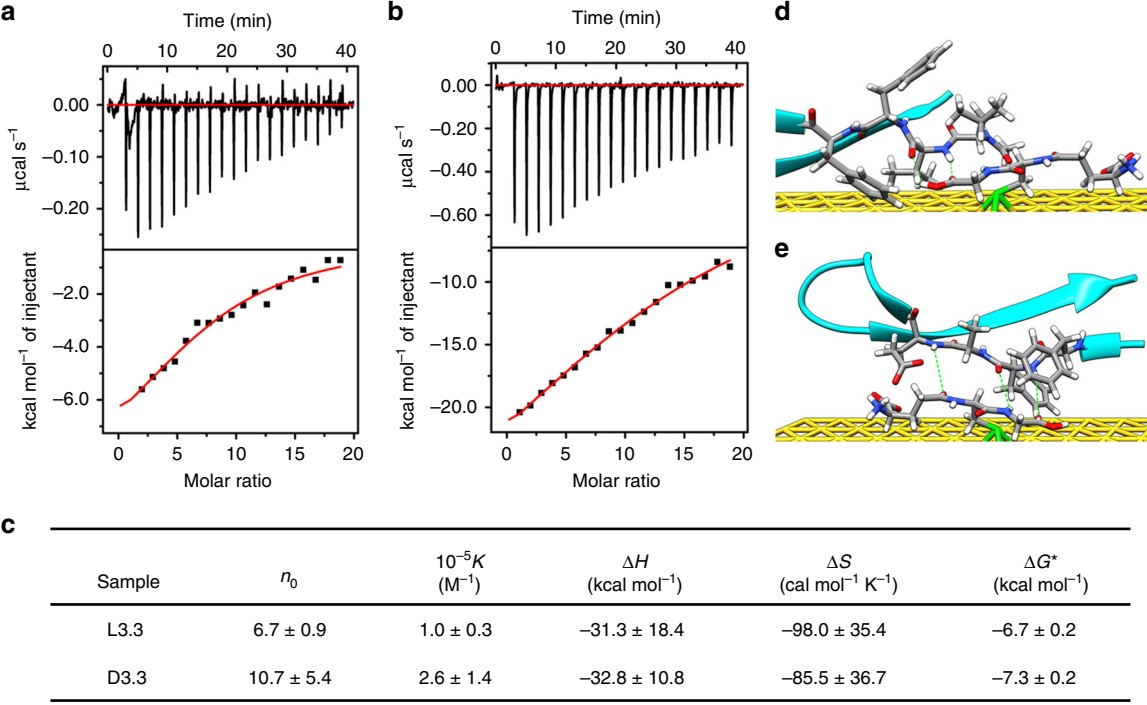

**Fig. 2 Analysis of interaction between L3.3/D3.3 and Aβ42 monomer. a, b** Typical ITC data for titration of Aβ42 (0.138 mM) into solution of L3.3 (1.5 μM) (**a**) and D3.3 (1.5 μM) (**b**) at 25 °C. (Upper) raw data and (lower) integrated data. The solid curves represent the best-fit results using a one-site binding model. The experiment was repeated for three times. **c** Summary of binding stoichiometry $n_0$, binding constant $K$, enthalpy changes $\Delta H$, entropy changes $\Delta S$, and Gibbs free energy changes $\Delta G$ of binding process of Aβ42 monomer on L3.3 (**a**) and D3.3 (**b**). Error bars indicate mean ± s.d. *$P < 0.05$, two-sided Student's $t$ test. For detailed statistical analysis see Supplementary Table 7. **d, e** The most stable structures of Aβ17-36 with L- (**d**) and D-GSH-coated Au (111) surface (**e**) obtained from molecular docking simulation (atom color: C, gray; H, white; N, blue; O, red; S, green; Au, yellow). Green dotted lines represent hydrogen bonding. Source data are provided as a Source Data file.

the Gibbs free energy change $\Delta G$ of binding process of Aβ42 monomer on D3.3 is significantly lower than that on L3.3 ($-7.3 \pm 0.2$ vs. $-6.7 \pm 0.2$ kcal mol$^{-1}$, *$P < 0.05$), indicating the binding process of Aβ42 with D3.3 is more energetically favorable. Hence, the strong affinity between D3.3 and Aβ42 monomer would substantially lower the concentration of free Aβ42 monomers, shift the equilibrium away from Aβ42 aggregation, and thus hinder fibril growth[36].

The theoretical simulation was further conducted to elucidate the binding mode of Aβ peptides onto the surfaces of L3.3 or D3.3. Based on Supplementary Fig. 12a model of Au (111) layers decorated with one L- or D-GSH molecule via Au–S bond is adopted to represent L3.3/D3.3. Moreover, to simplify the calculation, the Aβ17-36 fragment is used to mimic Aβ42 because it has a similar "beta-turn-beta" structure and assembly behavior (Supplementary Fig. 13a)[38]. The L- or D-GSH stabilized Au (111) layers are optimized through density functional theory (DFT) calculation[39–41] (Supplementary Fig. 13b, c), which are subsequently docked with Aβ17-36 peptide to figure out the most stable binding conformation[42] (Fig. 2d, e and Supplementary Fig. 13d, e). The corresponding binding free energy of Aβ17-36 with D-GSH stabilized Au (111) layer is $-4.64$ kcal mol$^{-1}$, larger than that with L-GSH stabilized Au (111) layer ($-4.19$ kcal mol$^{-1}$), confirming that D3.3 has stronger interaction with Aβ than L3.3. This simulation result is consistent with binding constant measurement (Fig. 2c) and inhibition activity observation (Fig. 1a). Most importantly, as demonstrated in Fig. 2d, e, the L- or D-GSH on Au (111) layers prefer interacting with the KLVFFA segment of Aβ via hydrogen bonding, which is the widely accepted key element for forming Aβ aggregation[43]. This result reveals why

D3.3 and L3.3 possess a good target capability to prevent Aβ aggregation.

**Inhibition effect on Aβ-mediated cellular toxicity.** The ability of L3.3 or D3.3 to inhibit Aβ42 aggregation holds the promise of protecting cells from Aβ-mediated toxicity. So, the Cell Counting Kit-8 (CCK-8) assay was performed to probe the cell viability of human neuroblastoma cell line SH-SY5Y. As a control, CCK-8 assay is carried out with 25–500 nM of L3.3 or D3.3, and the result displays that both L3.3 and D3.3 have almost no cytotoxicity to cells, indicated by >80% cell viability (Supplementary Fig. 14a). On the contrary, pure Aβ42 is quite toxic to the SH-SY5Y cells, leading to <45% cell viability (Fig. 3a). When L3.3 or D3.3 is added to Aβ42, SH-SY5Y cells are saved from Aβ-mediated toxicity. Especially, D3.3 improves the cellular viability of up to 80% while L3.3 achieves survival of 72%.

Moreover, terminal transferase dUTP nick end labeling (TUNEL) assay was conducted to detect cell apoptosis. The normal cell nuclei appear blue while the apoptotic cell nuclei show green with TUNEL-4′,6-diamidino-2-phenylindole (DAPI) co-staining (Fig. 3b). The D3.3 treatment causes a more obvious decrease in SH-SY5Y cell apoptosis than L3.3. To quantify cell apoptosis, SH-SY5Y cells treated with Aβ42 plus D3.3 (or Aβ42 plus L3.3) for 48 h were stained with annexin V-fluorescein isothiocyanate (Annexin V-FITC)/propidium iodide (PI; Supplementary Fig. 14b). Compared with pure Aβ42, incubation of Aβ42 with L3.3 or D3.3 gives rise to a decrease of apoptosis from 42.9% to 36.2% or 32.1% (Fig. 3c, d). Altogether, both L3.3 and D3.3 can protect SH-SY5Y neuronal cells from cytotoxicity induced by Aβ42 aggregation, and D3.3 has a more significant

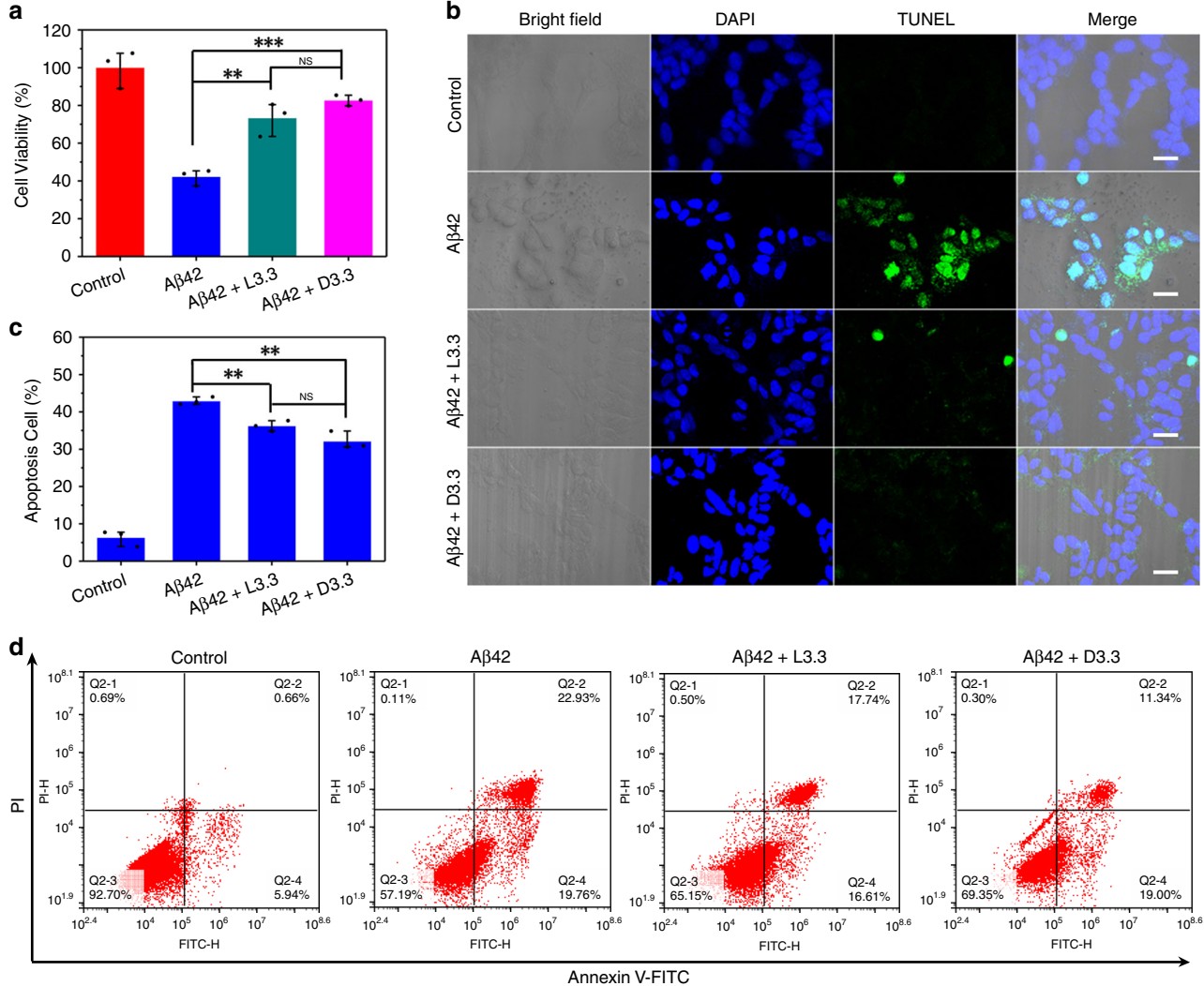

**Fig. 3 Inhibition effect of L3.3 and D3.3 on Aβ42-induced neuronal death. a** Cell toxicity of Aβ42 (40 μM) in the absence and presence of L3.3 and D3.3 (110 nM). **b** Bright-field and confocal fluorescence images of SH-SY5Y cells treated with Aβ42 (40 μM) in the absence and presence of L3.3 and D3.3 (110 nM) for 48 h. Scale bars, 50 μm. **c** Population of apoptosis cells summarized from flow cytometry analysis. **d** Flow cytogram representing apoptosis assay based on annexin V-fluorescein isothiocyanate (Annexin V-FITC)/propidium iodide (PI) staining of SH-SY5Y cells after treatment with different therapeutic groups for 48 h. Error bars indicate the s.d. (n = 3 independent samples). **P < 0.01; ***P < 0.001; NS not significant; one-way ANOVA, followed by Holm-Sidak post hoc test. For detailed statistical analysis see Supplementary Tables 6 and 7. Source data are provided as a Source Data file.

effect in reducing the apoptosis rate due to its stronger interaction with Aβ42 and higher inhibition ability.

**Biodistribution and toxicity evaluation in vivo.** The therapeutic efficacy of L3.3 and D3.3 for AD strongly depends on their blood circulation time and ability to override the BBB. Previous study has demonstrated that there is high level of GSH transporters located in the brain[23], so it is expected that chiral GSH coating enables Au NPs of small sizes possessing more efficient BBB permeability. L3.3 or D3.3 (25 mg kg$^{-1}$) was injected intravenously into healthy Kun Ming (KM) mice, and their biodistribution was determined by measuring the amount of Au element through inductively coupled plasma mass spectrometry (Fig. 4a, b and Supplementary Fig. 15). As summarized in Fig. 4a, the amount of Au in the brain of L3.3- or D3.3-injected mice reaches the maximum concentration (10 times higher than that of the control group) at 12 h or 6 h, respectively. This result discloses that L3.3 and D3.3 could be transported from the blood circulation into brain. Notably, the surface chirality of Au NPs

significantly influences the biodistribution of D3.3 and L3.3 in the brain at first 6 h, implying that L3.3 and D3.3 may cross the BBB through transporter proteins[44] besides spontaneous penetration[19,45]. Moreover, the Au content begins to decrease after 12 h for L3.3-injected mice and 24 h for D3.3-injected mice, respectively, suggesting that the clearance of D3.3 from the brain is slower than that of L3.3. As expected, the amount of L3.3 and D3.3 in the blood gradually decreases with the prolonged time because of clearance and aggregation (Fig. 4b). The clearance of L3.3 and D3.3 in plasma follows the first-order elimination kinetics (Fig. 4c). The half-life of L3.3 and D3.3 is 14.3 h and 14.8 h, respectively, which is close to the reported values of other Au NPs[45].

To evaluate the biosafety of L3.3 and D3.3 for AD treatments, serum biochemistry assay and hematoxylin and eosin (H&E) staining of major organs were performed. The results display that the hematological parameters first suffer slight fluctuation at the early stage of injection (6 h and 12 h), then recover to the normal range (from 24 h to 48 h; Fig. 4d and Supplementary Fig. 16). The histological section of heart, liver, spleen, lung, and kidney with

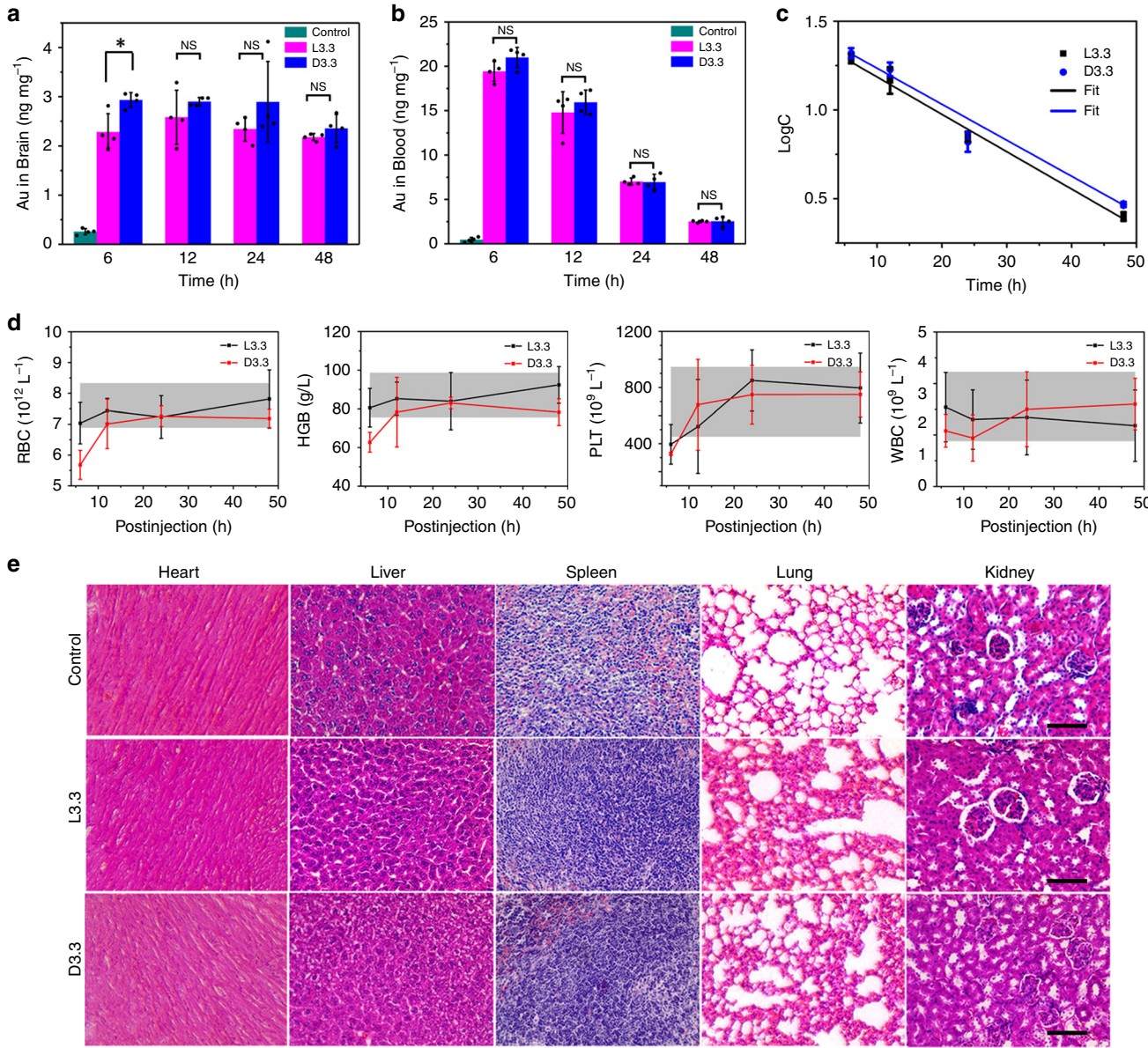

**Fig. 4 Biodistribution and toxicity evaluation of L3.3 and D3.3 in vivo. a, b** Biodistribution of L3.3 and D3.3 in the brain (**a**) and blood (**b**) at 6 h, 12 h, 24 h, and 48 h post-injection. Error bars indicate the s.d. ($n = 4$ mice per group). *$P < 0.05$; NS not significant; two-sided Student's $t$ test. For detailed statistical analysis see Supplementary Tables 6 and 7. **c** Elimination kinetics of L3.3 and D3.3 in plasma. The concentration-time profiles of L3.3 and D3.3 in plasma (**b**) fit the first-order elimination. The fitting equation for L3.3 and D3.3 groups is $y = 1.396 - 0.0211x$ ($r^2 = 0.992$) and $y = 1.43758 - 0.0203x$ ($r^2 = 0.989$), respectively. **d** Hematological parameters, including red blood cells (RBC), hemoglobin (HGB), platelets (PLT), and white blood cells (WBC) of the KM mice after treatment with L3.3 or D3.3 for 6 h, 12 h, 24 h, and 48 h. The shadow regions represent the normal range of control groups. Error bars indicate the s.d. ($n = 3$ mice per group). For detailed statistical analysis see Supplementary Table 6. **e** H&E staining assays of the heart, liver, spleen, lung, and kidney from all experimental groups ($n = 2$ mice per group). Scale bars, 100 μm. Source data are provided as a Source Data file.

H&E staining after 48 h injection of L3.3 or D3.3 reveals that neither noticeable organ damage nor inflammation is distinguished (Fig. 4e). In sum, at least partially if not all, the in vivo administration of L3.3 and D3.3 is safe under the current dose.

**Rescuing memory deficits in AD model mice.** The high inhibition effect of L3.3 and D3.3 on Aβ aggregation and Aβ-mediated cellular toxicity, in addition to their good BBB permeability and biocompatibility, encourage us to investigate in vivo therapeutic efficiency for AD. APPswe/PS1-dE9 [(amyloid precursor protein/presenilin protein 1 (APP/PS1)] double-transgenic mouse model, which is characteristic of Aβ plaque deposition,

gliosis, and cognitive impairment[46], is adopted to check in vivo therapeutic efficiency of chiral Au NPs for AD. The model mice were weekly intravenously administrated for 4 weeks with L3.3 or D3.3. Subsequently, the spatial cognition and memory of the model mice were assessed using the Morris water maze (MWM) test. The latency of different group mice for searching the hidden platform was measured daily for 5 days (Fig. 5a), and the probe trial was conducted after removal of the hidden platform on the 6th day (Fig. 5b–d). Compared with the wild-type (WT) mice, the AD model mice show the obvious deficits, as evidenced by longer escape latency time (Fig. 5a), shorter time spent in the target quadrant after removal of the platform (Fig. 5b), less platform crossing (Fig. 5c) and more random motion paths (Fig. 5d).

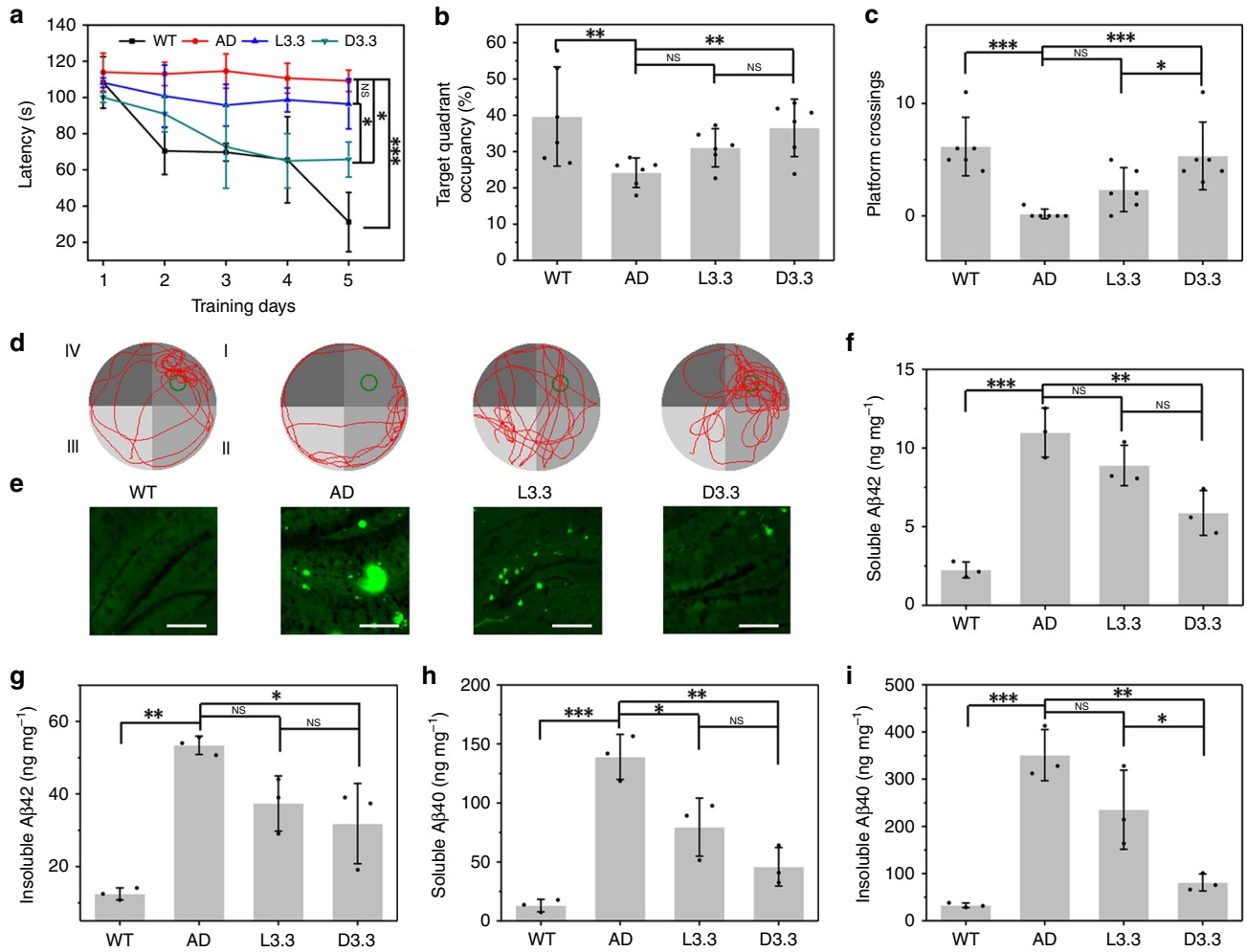

**Fig. 5 Effect of L3.3 and D3.3 on rescuing memory impairments in AD model mice. a** Latency for escape to platform in the training phase. **b** Percentage (%) of time spent in the target quadrant in probe test. **c** Number of crossing platform time in probe test. Error bars indicate the s.d. ($n = 6$ mice per group). *$P < 0.05$; **$P < 0.01$; ***$P < 0.001$; NS not significant; one-way ANOVA, followed by Holm-Sidak post hoc test. For detailed statistical analysis see Supplementary Tables 6 and 7. **d** Track of probe test. **e** Immunofluorescence for Aβ in the hippocampus of WT control mice, AD control mice, AD mice with L3.3 or D3.3 treatment. Scale bars, 200 μm. **f–i** Soluble Aβ42 (**f**), insoluble Aβ42 (**g**), soluble Aβ40 (**h**), and insoluble Aβ40 (**i**) in the brains measured by ELISA. Error bars indicate the s.d. ($n = 3$ mice per group). *$P < 0.05$; **$P < 0.01$; ***$P < 0.001$; NS not significant; one-way ANOVA, followed by Holm-Sidak post hoc test. For detailed statistical analysis see Supplementary Tables 6 and 7. Source data are provided as a Source Data file.

During the 5-day training and subsequent probe trial, the animals treated with D3.3 display remarkable improvement in spatial learning and memory in comparison with AD model mice, while L3.3 treatment does not cause significant improvement (Fig. 5a–d and Supplementary Movie 1–8). Specifically, the animals administrated with D3.3 exhibit shorter latency to escape onto the platform, slightly longer swimming time at the platform location, more platform crossing and better-focused search strategies for the platform in the quadrant compared with those treated with L3.3 (Fig. 5a–d).

After the MWM test, the mice were killed and the immunofluorescence analysis was carried out to evaluate the ability of L3.3 and D3.3 on reducing Aβ deposition in the brains. A large amount of Aβ deposition is found in the hippocampus of AD mice (Fig. 5e). Impressively, both D3.3 and L3.3 treatment decrease Aβ deposition, but D3.3 treatment is more efficient than L3.3 (Fig. 5e). The Aβ level in the hippocampus of mice is further quantitatively measured by an enzyme-linked immune sorbent assay (ELISA)[47]. The amount of soluble and insoluble Aβ42, as well as Aβ40, are significantly decreased by D3.3 treatment (Fig. 5f–i). Notably, L3.3 treatment leads to a significant

difference in the number of soluble Aβ40, but this decrease is not intense enough to cause obvious improvement in the behavior of AD mice. In addition, during the therapeutic period, the body weight of mice is not significantly affected by various treatments (Supplementary Fig. 17). Histopathology assay proves that no significant difference is detected in pathological signs between the treated group and the control group (Supplementary Fig. 18). Taken together, a 4-week weekly treatment with D3.3 significantly rescues the spatial learning and memory impairments in AD mice via reducing Aβ deposition, presenting a promising therapeutic strategy for AD.

## Discussion

We demonstrate the unique advantages of chiral molecule stabilized NPs for AD treatment. Both theory and experiment disclose that compared with pure L- and D-GSH molecules, L- and D-GSH stabilized Au NPs show much-improved prevention of Aβ aggregation via adsorption of peptide monomers on their curvature surfaces (Supplementary Figs. 1 and 8), and the NPs with the size of 3–4 nm possess the best inhibition efficiency and

protective effect of neurons against Aβ42 aggregates-induced cellular toxicity. In vivo mice experiment via the intravenous injection demonstrates that L3.3 and D3.3 can be transported from the blood circulation into the brain, and D3.3 has higher brain distribution. Most importantly, a 4-week weekly intravenous administration of D3.3 decreases Aβ plaque deposition in the brain and rescues the memory deficits of AD mice. It is worth pointing out that regardless of treatment method or rescue effect, our chiral Au NPs with small size exhibit obvious advantage over the current reported NP systems (Supplementary Table 5). In detail, our chiral NPs can cross the BBB through simple intravenous injection. We notice that the currently reported NP systems are either through stereotactic brain injection or intravenous injection with the assistance of cyclosporine A to modulate P-glycoprotein function. The operation of stereotactic brain injection is complicated and the mortality rate is high. Besides, modulating the P-glycoprotein function to improve cerebral uptake has severe adverse effects that include an increased risk for serious infections. The last but not the least, the target quadrant occupancy of D3.3-treated AD mice is closest to that of the WT group among all the reported NPs (up to 94.9%). The Aβ-based drugs suffer failures to date, likely in part because the drugs tested lack sufficient target engagement or have toxic effects. We highly expect that chiral nanomedicine will open a potential era toward the treatment of Alzheimer's disease thanks to their size- and stereo-determined inhibition of peptide or protein aggregation, the capability of crossing the blood–brain barrier, good biosafety and rather rapid clearance from the body.

## Methods

**Materials.** Aβ42 was purchased from CS Bio Co (USA). D-GSH (≥98%) was bought from GL Biochem., Ltd. (Shanghai, China). L-GSH (≥98%), sodium citrate tribasic dehydrate (≥99%), gold (III) chloride trihydrate (HAuCl$_4$·3H$_2$O), Thioflavin T (ThT), sodium borohydride (NaBH$_4$), 1,1,1,3,3,3-hexafluoro-2-propanol (HFIP), 2M NaOH aqueous solution, phosphotungstic acid, and PBS (pH = 7.4) were got from Alfa Aesar. Dimethyl sulfoxide (DMSO) and acetonitrile were purchased from Beijing Chemical Reagent Company. Concentrated hydrochloric acid, concentrated nitric acid, and hydrogen peroxide (H$_2$O$_2$) were of the highest purity and bought from Sigma Aldrich (USA). Standard solution of gold (1 mg mL$^{-1}$) was obtained from the National Analysis Center for Iron and Steel (China). CCK-8 assay was purchased from Dojindo Molecular Technologies, Inc. (Kumamoto, Japan). TUNEL assay kit and DAPI were obtained from Roche Applied Science (Roche, Mannheim, Germany). Annexin V-FITC/PI staining assay was bought from Beijing Dingguo Changsheng Biotechnology Co., Ltd. (Genview, China). KM mice and C57BL/6 mice (WT mice) were got from Beijing Vital River Laboratory Animal Technology Co., Ltd. (Beijing, China). The APPswe/PS1dE9 transgenic mice (AD mice) were purchased from the Nanjing Biomedical Research Institute of Nanjing University (Nanjing, China). All the animal experiments were approved by an ethical committee of National Center for Nanoscience and Technology, Harbin Institute of Technology and College of Pharmacy of Harbin Medical University. All ethical obligations were complied with. Mice were housed with water and food ad libitum and with 12 h/12 h light/dark cycle, under standard environmental conditions (temperature and humidity). The β-amyloid (D3D2N) primary antibody and the anti-mouse Alexa Fluor 488 secondary antibody for immunofluorescence analysis were purchased from Cell Signaling Technology (Beverly, MA, USA). Mouse Aβ1-42 ELISA Kit (MG2509) and Mouse Aβ ELISA Kit (MG10283) were achieved from the Trust Specialty Zeal Biological Trade Co., Ltd. (USA). All other chemicals and reagents were of analytical grade and used without further purification. Milli-Q water (electrical conductivity ~ 18 MΩ cm$^{-1}$) was used to prepare solutions.

**Instruments.** The hydrodynamic diameter and surface charge of as-synthesized Au NPs were determined by dynamic light scattering (DLS) using a Zetasizer Nano ZS instrument (Malvern Instruments, UK). TEM images were acquired with a Tecnai G2 F20 U-TWIN (FEI Company, USA) and Tecnai G2 20 S-TWIN (FEI Company, USA) under the accelerating voltage of 200 kV. UV-vis spectra were recorded on a Hitachi U-3900 spectrophotometer (Hitachi Corportion, Tokyo, Japan). The size of nanoparticles was analyzed using the Nano Measurer 1.2 software. Fourier transform infrared (FTIR) spectra were obtained on a Perkin Elmer Spectrum One spectrometer by using the KBr disk method. CD spectra were achieved by a Jasco J-1500 spectropolarimeter in aqueous solution. The X-ray diffraction (XRD) patterns were recorded on a D/MAX-TTRIII (CBO) instrument with Cu Kα radiation (λ = 1.542 Å) operating at 50 kV and 300 mA.

**Monte Carlo simulation.** Monte Carlo simulation was performed under the NVT ensemble, which meant constant particle number, volume, and temperature. The length of the cubic box was 38.7 nm, and the periodic boundary condition was applied in all the three dimensions. The NPs were distributed in the system randomly, and the positions of NPs in the system were fixed. The peptide chains were moved by following pivot, crankshaft, kink-jump, and translation methods[48]. Peptide chains with a number of 1330 were included in the system. In all, $1 \times 10^4$ Monte Carlo steps under a thermal condition were firstly performed for relaxation of the peptide chains, $1 \times 10^6$ Monte Carlo steps were then performed for the production run, and the last $1 \times 10^5$ steps were used for data analysis. The judgment for forming or breaking hydrogen bonds was made before each Monte Carlo move of peptide bead. For each system, 10 samples with different initial coordinates were performed to achieve the average.

**Preparation of L3.3 or D3.3.** L3.3 or D3.3 was synthesized by the modified Kumar's method[29]. Typically, an aqueous solution of HAuCl$_4$·3H$_2$O (84 μL, 476.3 mM) was added to 5 mL fresh L- or D-GSH aqueous solution (0.12 mmol), which was then poured into 36 mL H$_2$O. After continuous stirring for half an hour, 2 mL NaBH$_4$ aqueous solution (0.88 mmol) was added slowly with stirring to fully reduce the gold ions, and then the solution was stirred for another 2 h until the solution became a brown color. The resultant product was collected by ultrafiltration at $10,620 \times g$ for 15 min and then washed three times with deionized water to remove free GSH molecules. Finally, the resulting L3.3 or D3.3 was dispersed in aqueous solution and stored at 4 °C before use in further study.

**Preparation of L9 or D9.** L9 or D9 was synthesized by the modified method[30]. Typically, an aqueous solution of HAuCl$_4$·3H$_2$O (0.125 mL, 476.83 mM) was mixed with an aqueous solution of L- or D-GSH (0.15 mM, 100 mL). After stirred for half an hour, the pH value of the solution was tuned to 11.00 using 2 M NaOH aqueous solution. Afterward, the mixture was kept overnight at 37 °C. The free GSH molecules in solution were removed by centrifugation at $10,620 \times g$ for 15 min, and then the precipitate was dispersed in the aqueous solution before use.

**Preparation of L15 or D15.** L15 or D15 was synthesized by the modified method[30]. Typically, an aqueous solution of HAuCl$_4$·3H$_2$O (0.125 mL, 476.83 mM) was mixed with an aqueous solution of L- or D-GSH (0.15 mM, 100 mL). After stirred for 3 min, the pH value of the solution was tuned to 11.00 using 2 M NaOH aqueous solution. Afterward, the mixture was kept overnight at 37 °C. The free GSH molecules in solution were removed by centrifugation at $10,620 \times g$ for 10 min, and then the precipitate was washed three times and finally dispersed in the aqueous solution before use.

**Preparation of C3.5.** For comparison, C3.5 was also prepared according to the literature[31]. Typically, an aqueous solution of HAuCl$_4$·3H$_2$O (1%, 1 mL) and sodium citrate solution (0.03 M, 2 mL) were added to 50 mL of purified water and stirred. Then freshly prepared KBH$_4$ aqueous solution (0.1 M, 1 mL) was added, and the solution color changed from colorless to wine red. The solution was left undisturbed for 2 h. The obtained citrate-capped Au nanoparticles were stored at 4 °C.

**Preparation of Aβ42 solution.** In total, 1 mg Aβ42 peptide was first dissolved in 1 mL of HFIP and shaken at room temperature for 24 h in a sealed vial. Afterward, HFIP was removed by evaporation under a gentle stream of nitrogen gas. Subsequently, an aliquot of Aβ42 was redissolved with 120 μL of DMSO and diluted in 5.5 mL of pure water. The final concentration of Aβ42 was ~40 μM. The Aβ42 solution was filtered through a 0.22-μm filter before injected on a Superdex 200 10/300 GL column (GE HealthCare) equilibrated in PBS buffer (10 mM, pH 7.4) at a flow rate of 0.5 mL min$^{-1}$. Absorbance at 280 nm was recorded (Wyatt Technology, CA, USA). The oligomer A11 polyclonal antibody (Thermo Fisher, 1:250 dilution) was used according to the manufacturer's instruction. A 10 μL of each sample was applied onto untreated nitrocellulose membranes and allowed to dry. A HRP-conjugated goat anti-rabbit IgG antibody (Invitrogen, Carlsbad, CA, 1:3000 dilution) was used to detect bound A11 antibodies using a chemiluminescence with the Clarity™ ECL Western Blotting Substrate (Bio-Rad).

**Fibrillation experiments.** The fibrillation of Aβ42 (40 μM) in the absence and presence of various types of Au NPs were incubated at 37 °C in a shaker. The concentration of L3.3/D3.3, L9/D9, and L15/D15 were 110 nM, 15 nM, and 5 nM, respectively, to keep the same surface area in each group. The Aβ42 fibrillation process was monitored by ThT fluorescence and CD spectroscopy at specific time points.

**ThT fluorescence assay.** The mixture solution of 100 μL Aβ42 sample and ThT solution (10 μL and 20 μM) was incubated in a 96-well plate at room temperature for 15 min. Fluorescence was measured with the microplate reader (EL800, Bio-Tek Instrument, USA). Excitation and emission wavelengths were 450 and 482 nm, respectively. The experiments were performed in triplicate.

**CD studies and secondary structure analysis**. CD spectra were employed for investigating the inhibition effect of chiral Au NPs on Aβ42 fibrillization, which were recorded by the Jasco J-1500 spectropolarimeter (Jasco Co. Ltd., Tokyo, Japan). The experiments were performed by infusing a 200-μL sample into a quartz cell with a path length of 1 mm. The experimental parameter was set as follows: scan wavelength, 190–260 nm; scan speed, 500 nm $min^{-1}$; bandwidth, 1 nm. To avoid the interference of GSH-coated Au NPs, the spectra of the mixed Au NPs and Aβ42 samples were collected using the CD spectra of GSH-coated Au NPs alone as the baseline. It should be noted that in the CD experiments, the solubilizing assistant DMSO for Aβ42 was replaced by acetonitrile. The protein secondary structure was analyzed using the CDPro software.

**TEM characterization**. To prepare the TEM specimen, a 10 μL sample was dropped onto a carbon-coated 300-mesh copper grid. After 15 min, the remaining solution was soaked up from the edge of the grid using filter paper. Afterward, the specimen was stained with 10 μL of 1.5% (w/v) phosphotungstic acid solution (pH = 7.4) for 2 min, and then the staining solution was drawn away from the edge of the grid with filter paper. The grid was washed with 10 μL of deionized water three times and dried at room temperature. Finally, the sample was observed using Tecnai G2 20 S-TWIN (FEI Company, USA) under the accelerating voltage of 200 kV.

**AFM imaging**. AFM morphology images were recorded using Multimode-8 AFM (Bruker, USA). To prepare a specimen for AFM, a 10 μL sample was deposited onto freshly cleaved mica surface for 10 min. The sample was then briefly rinsed with Milli-Q water and dried with a gentle stream of nitrogen gas. All morphology images were recorded using an intelligent mode with a $512 \times 512$ pixel resolution and a scan speed of 1.0–1.5 Hz. Analysis of the images was carried out using the NanoScope Analysis 1.40 Software.

**ITC measurements**. ITC experiment was performed in a MicroCal ITC200 instrument (Malvern, Sweden) at 25 °C and the stirring speed was set to 750 rpm. In the ITC experiment, 40 μL of Aβ42 solution (138 μM) was injected in each step of 2 μL into 200 μL of L3.3/D3.3 solution (1.5 μM). The experimental thermogram was fitted to a one-site binding model implemented in the instrument software supplied by the manufacturer.

**DFT computation and molecular docking**. The $(1 \times 3)$ supercell model with a three-layered periodic slab separated by a vacuum region of 15 Å was used to model the Au (111) surface and its interaction with ligands (L-GSH or D-GSH). All calculations were performed using the Vienna ab initio simulation package (VASP) with plane-wave pseudopotential method[39]. The electronic exchange and correlation effects were described by the Perdew-Burke-Ernzerhof (PBE) functional[40] with the generalized gradient approximation (GGA) and the core electrons were described by the full-potential projector augmented wave (PAW) method[41]. An energy cutoff of 400 eV for the plane-wave expansion was used and the force on the relaxed atoms was $<-0.03$ eV $Å^{-1}$. The spin-polarization was taken into account in all calculations. The Brillouin zone was sampled by $(1 \times 3 \times 1)$ k-point grid generated within the Monkhorst-Pack scheme.

The molecular docking simulation was carried out by using Auto-Dock 4.2.6 and AutoDockTools 1.5.6 was employed to generate the docking input files and analyze the docking results[42]. To identify potential binding sites of Aβ17-36 on L- or D-GSH stabilized Au (111), a big grid box size of $126 \times 126 \times 126$ points with a large spacing of 0.603 Å between the grid points was implemented and the grid box was large enough to cover the entire surface of Au (111). The ones with the lowest binding energy were selected for detailed analysis and further studies. The affinity map of the adsorbed ligand on Au (111) and the peptide was calculated using Auto Grid. Lamarckian Genetic Algorithm (LGA) added a local minimization to the genetic algorithm, enabling modification of the gene population. The docking parameter was as follows: trials of 100 dockings, the population size of 150, the random starting position and conformation, the mutation rate of 0.02, the cross overrate of 0.8, the local search rate of 0.06, and 25 million energy evaluations. Final docked conformation was clustered using a tolerance of 2.0 Å root-mean-square deviations (RMSD).

**Cell viability assay**. To verify that the L3.3 or D3.3 could decrease the cell mortality induced by Aβ aggregation, SH-SY5Y cells were plated at a concentration of $1.0 \times 10^4$ cells $well^{-1}$ in 96-well plates with 200 μL of media and incubated overnight. Aβ42 monomers (40 μM) alone or together with L3.3 or D3.3 (110 nM) were dispensed into the SH-SY5Y cells, and the cells were further incubated for 48 h at 37 °C and in 5% $CO_2$ incubator. Cells were treated with the same amount of PBS as a control. Cytotoxicity was measured by using a CCK-8 assay. A 10 μL of CCK-8 solution was added to each well and the absorbance at 450 nm was measured after 1 h of incubation using a microplate reader (EL800, Bio-Tek Instrument, USA). To further investigate the cytotoxicity of L3.3 and D3.3, the various concentration of L3.3 or D3.3 was dispensed into the SH-SY5Y cells, and further incubated for 48 h at 37 °C and in 5% $CO_2$ incubator. Cytotoxicity was measured using a CCK-8 assay.

**TUNEL-DAPI co-staining assay**. DNA fragmentation induced by Aβ42 fibers was examined by using an in situ cell death detection kit following the manufacturer's protocol. Briefly, cells were cultured with Aβ42 peptides (40 μM) and L3.3/D3.3 (110 nM) at 37 °C for 48 h, meanwhile, cells were treated with 40 μM of Aβ42 or the same amount of PBS buffer as a control. After that, cells in chamber slides were fixed with 4% formaldehyde for 15 min and permeabilized with 0.1% Triton X-100 in PBS. The cells were cultured with TUNEL reaction mixture for 1 h. For nuclear staining, cells were incubated with 1 μg $mL^{-1}$ of DAPI for 10 min at 37 °C. At the end of incubation, the cells were rinsed with PBS and the images were captured using a laser confocal microscope (Zeiss 710, Zeiss, Oberkochen, Germany).

**Flow cytometry analysis**. The externalization of phosphatidylserine to the cell surface was detected by flow cytometry using a double-staining with Annexin V-FITC and PI. The same cell lines were cultivated in similar conditions for TUNEL staining. After 48 h, cells were harvested by trypsinization and adjusted at $2.0 \times 10^5$ cells $mL^{-1}$ with combining buffer. The cell suspension was then incubated with 5 μL of 5 μM Annexin V-FITC solution and 10 μL of 20 μg $mL^{-1}$ PI solution at room temperature in the dark for 15 min. Data acquisition was then performed by a NovoCyte flow cytometer (ACEA Biosciences, Inc., San Diego, USA) using NovoExpress 1.0.2 software.

**Biodistribution and toxicity evaluation in vivo**. Healthy KM mice ($n = 18$) were intravenously injected with D3.3 and L3.3 at a dose of 25 mg $kg^{-1}$, respectively. For in vivo biodistribution evaluation, at the indicated time points, the healthy mice were euthanized ($n = 4$). The blood and major organs including the brain, liver, kidney, and spleen were collected and digested with the help of aqua regia and hot temperature to analyze the amount of Au by inductively coupled plasma mass spectrometry (NexION 300X, Perkin Elmer, USA). At the same time points, 20 μL of blood was collected to perform blood panel analysis (HF-3800, HANFANG Ltd., Jinan, China). At 48 h after injection, major organs including heart, liver, spleen, lung, and kidney ($n = 2$) were collected for investigating the histopathology change after H&E staining. The sections were observed, and photos were taken using an inverted phase contrast microscope (AMEX1200, Life Technologies, WA, USA).

**MWM test**. The C57BL/6 mice were regarded as WT mice. The AD mice were divided into three groups, AD group and two sample groups ($n = 6$). All mice were male, and 5-month-old, which were administered with PBS, L3.3 and D3.3 (25 mg $kg^{-1}$), respectively, every week for 4 weeks. All mice were trained and tested in a water maze with a diameter of 1.2 m (RD1101-MWM-G, Shanghai Mobiledatum Information Technology Co., Ltd, China). The maze was filled with water and drained every 2 days. The temperature of the water was maintained at $22 \pm 1$ °C. Before the task, the water was dyed with titanium dioxide. The platform (0.1 m in diameter) was immobilized to 2 cm under the water surface during the training period. The maze contained a mass of fixed visual cues on the walls. The performance of each mouse was recorded with a video-tracking system. In the first 5 days, the mice received three acquisition trials daily. Briefly, three starting points, excluding the one quadrant with a platform, were randomly selected in three daily trials. Each mouse facing the wall of the tank was set free into the water. If the mouse found the hidden platform successfully within 120 s, it was allowed to stay on the platform for 30 s. If the mouse failed to reach the platform within 120 s, it was guided to stay on the platform for 30 s. Upon completion of the daily three trials, mice were removed from the maze, towel dried, and then returned to the cages. On the 6th day, the spatial memory of each mouse was explored by a probe trial without the platform. Every trained mouse was released into the water at the right opposite position of the escape platform and was allowed to swim freely for 120 s. The time that each mouse spent in searching for the platform in the quadrant where the platform used to be (target quadrant), and the number of times that each mouse crossed the platform were recorded.

**Immunofluorescence staining**. After the MWM test, the mice were killed via cardiac perfusion with 0.9% saline, and their brains were fixed in 4% paraformaldehyde. The brain was then placed in 15 and 30% sucrose solution, followed by being cut into 30 μm slices using a low temperature thermostat (Leica, Germany). Each section was incubated with 10% normal sera, followed by overnight incubation with the D3D2N primary antibody (1:200 dilution) at 4 °C. Then, sections were incubated with the anti-mouse Alexa Fluor 488 secondary antibody (1:200 dilution) for 1.5 h at room temperature. Finally, the brain sections were imaged using an Olympus microscope with a BX51 digital camera (Olympus, Japan).

**Measurement of soluble and insoluble Aβ42 as well as Aβ40 in the brain**. Brain tissue (half of one brain, three mice per group) for ELISAs were mechanically homogenized in 10 vol of ice-cold guanidine buffer (pH = 8.0) and mixed for 3–4 h at room temperature as described[48]. Then the diluted brain homogenate with 1:10 by ice-cold casein buffer (0.25% casein, 0.05% sodium azide, 20 μg $mL^{-1}$ aprotinin, 5 mM EDTA, 10 μg $mL^{-1}$ leupeptin, and pH 8.0) was centrifuged under $25,660 \times g$ at 4 °C for 30 min (Auanti J-26XP, Bechman Coulter Inc., Fullerton, USA). The supernatant was removed for ELISA measurement of soluble Aβ42 and Aβ40 without further dilution. The homogenate pellet that remained after centrifugation

was added to 440 μL cold formic acid and the tube was kept in ice. Each sample was sonicated in ice continuously until the pellet dissolved. Then high-speed centrifugation at $28{,}450 \times g$ for 1 h at 4 °C was performed (Allegra 64R, Bechman Coulter Inc., San Diego, USA). Afterward, 105 μL supernatant was diluted into 2 mL of formic-acid neutralization buffer (1 M Tris base/0.5 M $Na_2HPO_4$/0.05% sodium azide) immersed in ice for ELISA measurement of insoluble Aβ42 and Aβ40. Mouse Aβ1-42 ELISA Kit (MG2509) and Mouse Aβ ELISA Kit (MG10283) were used following the manufacturer's protocol.

**Statistical analysis**. All the data were presented with the mean ± standard deviation (s.d.) and were analyzed using SPSS 19.0 statistical analysis software (SPSS, Chicago, IL, USA). A two-sided Student's $t$ test was used to assess the difference between the two groups. Difference among multiple groups was analyzed via one-way analysis of variance (ANOVA), followed by Holm-Sidak post hoc test with ***$P < 0.001$, **$P < 0.01$, *$P < 0.05$.

**Reporting summary**. Further information on research design is available in the Nature Research Reporting Summary linked to this article.

## Data availability

The source data underlying Figs. 1–5, Supplementary Figs. 1–12, and Supplementary Figs. 14–17 are provided as Source Data file with this paper. Any other data are available from the corresponding author upon reasonable request. Source data are provided with this paper. Source data are provided with this paper.

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

## Acknowledgements

This work was supported financially by the National Key Basic Research Program of China (grant no. 2014CB931801 and 2016YFA0200700), National Natural Science Foundation of China (grant no. 21890381, 21721002 and 21475029), Frontier Science Key Project of Chinese Academy of Sciences (grant no. QYZDJ-SSW-SLH038), K. C. Wong Education Foundation, and National Science Fund for Distinguished Young Scholars (grant no. 51825202). The authors thank National Center for Protein Sciences at Peking University in Beijing, China, for assistance with ITC experiment and Dr. Hui Li for help with collecting data.

## Author contributions

K.H., S.L., and Z.T. conceived the idea for this study. K.H. synthesized the samples and participated in characterization, modeling, and cell and animal experiments. J.L. participated in AFM characterization. D.W. participated in TEM characterization. K.H., H.W., and K.W. conducted the DFT computation and molecular docking. K.H., B.L., and X.S. carried out the Monte Carlo simulation. K.H., Q.H., and H.Y.W. performed the cell viability assay. K.H. and J.Z. performed the biodistribution and toxicity evaluation in vivo experiment. K.H., K.L., and J.A. carried out the MWM experiment and immunofluorescence staining. S.L. and Z.T. supervised the project. K.H., S.L., and Z.T. wrote the manuscript. Q.C. helped revise the manuscript. All authors commented on the data and the manuscript.

## Competing interests

The authors declare no competing interests.
