## [Peer Review File · Nature Communications]

Reviewers' comments:

Reviewer #2 (Remarks to the Author):

The authors have designed and synthesized novel nanoparticle constructs for inhibition of amyloid-beta aggregation in vitro and reduced Alzheimer's disease pathology and memory deficits in vivo. These nanoparticles consist of a gold base stabilized by chiral glutathione. The authors demonstrate efficacy of these particles within in vitro and cell culture assays as well as in an animal model. Thus, this work identifies a potential new therapeutic approach for inhibition of amyloid protein aggregation and Alzheimer's disease. Some modifications, however, should be considered to clarify interpretation of the results and strengthen the manuscript.

- The abstract identifies anti-aggregation approaches as "one of the most efficient strategies" for halting the progression of Alzheimer's disease. While this is one therapeutic strategy, it is questionable whether the literature boasts this approach as among the most efficient. In fact, the authors recognize the drawbacks of this approach within the literature review of the Introduction. This statement within the abstract might be toned down to better align with the Introduction.
- On page 6, the authors state that the "fluorescence quenching of ThT caused by Au NPs can be ignored under the experimental conditions". Additional explanation about why this quenching can be ignored and/or data supporting this claim should be included.
- In Figure 1, panels a and c, the data shown represent a single aggregation experiment performed in triplicate. Was this experiment performed only once? If not, the more comprehensive data should be presented in some way, such as the lag time and rate of aggregation presented within the supplemental figures that compare the effect of NP size. If so, it raises concerns about the rigor of the acquired data. Similar concerns follow for aggregation figures within the supplemental information.
- Also in Figure 1, an indication of significance within panels a and c would better support the discussion on pages 6-7. In particular, the discussion indicating that D3.3 is a superior inhibitor (over L3.3) is difficult to deduce from the figure. Similar concerns follow for aggregation figures within the supplemental information.
- Supplementary Figure 6 is difficult to interpret without inclusion of the control data.
- From Figure 2, the authors conclude that the binding constant, K , is higher for D3.3 vs L3.3. Given the reported values and error, it is difficult to believe that this difference is significant. Was a statistical analysis performed?
- In cell experiments, it is unclear what aggregation state of amyloid-beta is used in cell treatments. The authors state in the methods that monomers are added to the cells; however, the protocol defined in the methods section for preparation of amyloid-beta (dilution of DMSO-dissolved amyloid-beta into aqueous buffer) is similar to that used for oligomer formation. Also, this preparation is added simultaneously with NPs; if the added preparation is oligomeric the role of NPs in preventing aggregation becomes less clear.
- On page 11, several differences in the biodistribution of D3.3 and L3.3 are discussed. It would be useful if these differences were supported by statistical indications within Figure 4.
- On page 12, some "significant" differences in behavior for mice treated with D3.3 and L3.3 are discussed (latency, time within target quadrant, platform crossings). Again, it would be useful if these significant differences were indicated within the figures.
- Within the Conclusions, the authors discuss the superiority of their NPs over other NPs described in the literature. A table within the supplementary information is included to support this discussion. However, the discussion within the manuscript itself is limited. This argument would be much stronger if a more thorough discussion were included within the manuscript itself.
- Throughout the manuscript, a number of grammatical errors should be corrected.

Reviewer #3 (Remarks to the Author):

The manuscript by Hou and colleagues describe the synthesis, characterization and utility of GSH-AU nanoparticles for potential treatment of Alzheimer's disease. Overall, the manuscript lacks adequate statistical analyses for the data presented and thus over-interpretation of the data is presented in both the results and discussion sections.

Specific concerns:

1. Figure 1. The secondary structure calculations in panel C do not match the representative CD spectra presented in panel b. This needs to be reconciled. There appears to be an over-estimation of the turn and random structures in both L3.3 and D3.3 + Ab42 results. Finally, the line estimation for Ab42+/-C3.5 does not accurately reflect the data or images in panels d and e.
2. No evidence is given for the statement that "fluorescence quenching of ThT caused by AU-NPs can be ignored."
3. Figure 2a- The binding constants and delta G are not different statistically between L3.3 and D3.3, however the discussion in the results state they are. This needs fixed as does all subsequent discussion and implications of this data to the toxicity results.
4. Figure 3c/d- no error bars or statistics were performed for these analyses- was this an n=1 experiment, if so it needs to be repeated and statistics performed.
5. Figure 4 there are no relevant differences between Au in brain and blood between L3.3 and D3.3 yet the results sections states differences. Neither is relevant and should be removed.
6. Figure 5. Limited statistics were performed, data is over-interpreted. No differences in the L3.3 group from untreated mice and yet presented as differences. Should be fixed.
7. For the discussion, since all Abeta-therapies have failed in the clinic some discussion on comparison of previous therapies to this should be presented at the preclinical level- to justify the statement that this is a better treatment option

Reviewer #2 (Remarks to the Author):

General Comments: The authors have designed and synthesized novel nanoparticle constructs for inhibition of amyloid-beta aggregation *in vitro* and reduced Alzheimer's disease pathology and memory deficits *in vivo*. These nanoparticles consist of a gold base stabilized by chiral glutathione. The authors demonstrate efficacy of these particles within *in vitro* and cell culture assays as well as in an animal model. Thus, this work identifies a potential new therapeutic approach for inhibition of amyloid protein aggregation and Alzheimer's disease. Some modifications, however, should be considered to clarify interpretation of the results and strengthen the manuscript.

Response: We are thankful for the reviewer's careful reading and constructive suggestions, which help us to greatly improve the quality of this manuscript. We have revised our manuscript according to the reviewer's insightful suggestions as below.

Comment 1. The abstract identifies anti-aggregation approaches as "one of the most efficient strategies" for halting the progression of Alzheimer's disease. While this is one therapeutic strategy, it is questionable whether the literature boasts this approach as among the most efficient. In fact, the authors recognize the drawbacks of this approach within the literature review of the Introduction. This statement within the abstract might be toned down to better align with the Introduction.

Response: Thanks a lot for the reviewer's careful reading and insightful suggestion. In order to make the Abstract section better align with the Introduction, we have changed the sentence "among which preventing aggregation of amyloid beta (A β) peptides is recognized as one of the most efficient strategies" to "among which preventing aggregation of amyloid beta (A β) peptides is recognized as one promising strategy". (please see the yellow highlighted parts in Abstract on page 2 in the revised manuscript).

Comment 2. On page 6, the authors state that the "fluorescence quenching of ThT caused by Au NPs can be ignored under the experimental conditions". Additional explanation about why this quenching can be ignored and/or data supporting this claim should be included.

Response: Thanks a lot for the reviewer's kind reminding and we are sorry for our carelessness.

In the experiment, we have measured the fluorescence spectra of ThT aqueous solution (2 μM) in the absence and presence of L3.3 (110 nM) or D3.3 (110 nM). As shown in below Figure R1, the added L3.3 or D3.3 does not cause an obvious decrease on the fluorescence intensity of ThT solution. Thus, we conclude that the quenching raised by gold nanoparticles under the experimental conditions can be ignored. What's more, we have fully investigated the literatures, which support that the quenching effect of gold-based nanosystems on the fluorescence of ThT can be ignored. (*Small* **2012**, 8, 3631-3639; *Chem. Eur. J.* **2015**, 21, 829-835; *Theranostics* **2017**, 7, 2996-3006.)

Figure R1. Fluorescence spectra of ThT aqueous solution (2 μM) in the absence and presence of L3.3 (110 nM) or D3.3 (110 nM). The excitation wavelength was 430 nm.

Revision: We have added Figure R1 into the revised Supplementary Information as Supplementary Figure 9, and also cited the literature (*Small* **2012**, 8, 3631-3639) as “Considering the fact that the fluorescence quenching of ThT caused by Au NPs can be ignored under the experimental condition¹² (Supplementary Fig. 9)” in the revised manuscript. (please see the yellow highlighted part on the middle of page 6).

Comment 3. In Figure 1, panels a and c, the data shown represent a single aggregation experiment performed in triplicate. Was this experiment performed only once? If not, the more comprehensive data should be presented in some way, such as the lag time and rate of aggregation presented within the supplemental figures that compare the effect of NP size. If so, it raises concerns about

the rigor of the acquired data. Similar concerns follow for aggregation figures within the supplemental information.

Response: Thanks a lot for the reviewer’s careful reading and insightful suggestion. The ThT experiment was performed for three times to ensure correctness of the result. Following the reviewer’s kind suggestion, we have comprehensively analyzed the lag time, time required to reach half of the maximum fluorescence intensity $t_{1/2}$ and apparent aggregation constant k of Figure 1a by fitting ThT data with sigmoidal curves. The same analysis on supplemental Figure 7a has been also conducted. The results are summarized in Table R1. It is clear that the chiral Au NPs exhibit inhibition effect on A β 42 aggregation by increasing the lag time and reducing the rate of A β 42 aggregation.

Table R1. Values of Lag time, time required to reach half of the maximum fluorescence intensity $t_{1/2}$ and apparent aggregation constant k obtained for the experiment shown in Figure 1a and Supplementary Figure 7a.

Sample	Lag time (h)	$t_{1/2}$ (h)	k (h ⁻¹)
A β 42	11.9	18.5 \pm 0.3	0.30 \pm 0.01
A β 42 + L3.3	20.9	37.4 \pm 5.4	0.12 \pm 0.02
A β 42 + D3.3	26.3	47.4 \pm 5.2	0.09 \pm 0.01
A β 42 + C3.5	12.0	31.7 \pm 7.6	0.10 \pm 0.03
A β 42 + D9	23.4	42.3 \pm 6.1	0.18 \pm 0.05
A β 42 + L9	16.1	27.1 \pm 1.7	0.10 \pm 0.02
A β 42 + D15	20.3	38.4 \pm 4.2	0.11 \pm 0.02
A β 42 + L15	15.6	21.9 \pm 0.8	0.32 \pm 0.08

Revision: We have added Table R1 into the revised Supplementary Information as Supplementary Table 4. and the corresponding discussions (To be more accurate, the lag time, time required to reach half of the maximum fluorescence intensity $t_{1/2}$ and apparent aggregation constant k are obtained by fitting ThT data with sigmoidal curves (Supplementary Table 4). It is clear that the chiral Au NPs exhibit inhibition effect on A β 42 aggregation by increasing the lag time and reducing the rate of A β 42 aggregation.) have been added in the revised manuscript (please see the

yellow highlighted part on page 6).

Comment 4. Also in Figure 1, an indication of significance within panels a and c would better support the discussion on pages 6-7. In particular, the discussion indicating that D3.3 is a superior inhibitor (over L3.3) is difficult to deduce from the figure. Similar concerns follow for aggregation figures within the supplemental information.

Response: Thanks a lot for the reviewer's careful reading and insightful suggestion. Following the reviewer's constructive suggestion, we have conducted statistics for panels a and c in Figure 1 as well as in Supplementary Figure 7. The results are summarized in Figure R2. With an indication of significance within panels in Figure R2 (Figure 1 and Supplementary Figure 7), it better supports the deduction that D-GSH-coated Au NPs are superior inhibitors over L-GSH-coated Au NPs.

Figure R2. a, ThT fluorescence assay of Aβ42 in the absence and presence of L3.3, D3.3 or C3.5. The fibrillation kinetics is fitted with a sigmoidal function. Error bars indicate the standard deviation (s.d.) (n = 3 independent samples). b, Analysis of protein secondary structure, Error bars indicate the s.d. (n = 3 independent samples). *P < 0.05, **P < 0.01, ***P < 0.001, Student's t-test. c, ThT fluorescence assay of Aβ42 in the absence and presence of L9, D9, L15 and D15. The fibrillation kinetics is fitted with a sigmoidal function. Error bars indicate the standard deviation

(s.d.) (n = 3 independent samples). **d**, Analysis of protein secondary structure. Error bars indicate s.d. (n = 3 independent samples). *P < 0.05, **P < 0.01, ***P < 0.001, Student's t-test.

Revision: We have replaced Figure 1a and Figure 1c with Figure R2a, Figure R2b, respectively. The information about statistical tests has been added in the figure legends (please see the yellow highlighted parts on page 34). Besides, Figure R2c and Figure R2d have been updated as Supplementary Figure S7a and Supplementary Figure S7c, respectively. The information about statistical tests also has been added in the figure legends (please see the yellow highlighted parts on page S14 in Supplementary Information).

Comment 5. Supplementary Figure 6 is difficult to interpret without inclusion of the control data.

Response: Thanks a lot for the reviewer's careful reading and insightful suggestion. We have added the control group and redrawn the Supplementary Figure 6. The new figure is shown below. Exactly as the reviewer commented, with inclusion of the control data, the discussion about Supplementary Figure 6 is easy to interpret.

Figure R3. Effect of L9, D9, L15, and D15 on Aβ42 fibrillization *in vitro*. **a**, ThT fluorescence

assay of A β 42 in the absence and presence of L9, D9, L15, and D15. The fibrillation kinetics is fitted with a sigmoidal function. Error bars indicate the standard deviation (s.d.) (n = 3 independent samples). *P < 0.05, ***P < 0.001, Student's t-test. **b**, CD spectra of A β 42 (40 μ M) in the absence and presence of L9, D9, L15 or D15 after co-incubation for 48 h. **c**, Analysis of protein secondary structure. Error bars indicate s.d. (n = 3 independent samples). *P < 0.05, **P < 0.01, ***P < 0.001, Student's t-test. **d**, AFM images of A β 42 (40 μ M) in the absence and presence of L9, D9, L15 or D15 after co-incubation for 48 h. **e**, TEM images of A β 42 (40 μ M) in the absence and presence of L9, D9, L15 or D15 after co-incubation for 48 h.

Revision: The original Supplementary Figure 6 has been replaced by Figure R3 and numbered as Supplementary Figure 7 in the revised Supplementary Information (please see page S14).

Comment 6. From Figure 2, the authors conclude that the binding constant, K , is higher for D3.3 vs L3.3. Given the reported values and error, it is difficult to believe that this difference is significant. Was a statistical analysis performed?

Response: Thanks a lot for the reviewer's careful reading and insightful suggestion. We are sorry for our carelessness. The error bars that we gave in the previous manuscript represented the fitting errors in one experiment. Following the reviewer's suggestion, we have repeated the ITC experiment for titration of A β 42 into the solution of L3.3 (Figure R4a) and D3.3 (Figure R4b) for three times. The mean values and error bars for binding stoichiometry n_0 , binding constant K , enthalpy changes ΔH , entropy changes ΔS and Gibbs free energy changes ΔG are summarized in Figure R4c. The statistical analysis is also conducted. The results indicate that the binding constant K between A β 42 and D3.3 is about 2.5 times larger than that between A β 42 and L3.3. Furthermore, the Gibbs free energy change ΔG of binding process of A β 42 monomer on L3.3 and D3.3 exhibits a significant difference (-6.7 ± 0.2 vs. -7.3 ± 0.2 kcal mol $^{-1}$, *P < 0.05), which indicates that the binding process of A β 42 with D3.3 is more energetically favorable.

Figure R4. Analysis of interaction between L3.3/D3.3 and A β 42 monomer. a,b, ITC data for titration of A β 42 (0.138 mM) into solution of L3.3 (1.5 μ M) (a) and D3.3 (1.5 μ M) (b) at 25°C. (Upper) raw data and (lower) integrated data. The solid curves represent the best-fit results using a one-site binding model. The experiment was repeated for three times. c, Summary of binding stoichiometry n_0 , binding constant K , enthalpy changes ΔH , entropy changes ΔS and Gibbs free energy changes ΔG of binding process of A β 42 monomer on L3.3 (a) and D3.3 (b). Error bars indicate mean \pm s.d.. * indicates $P < 0.05$, Student's t-test.

Revision: We have replaced the original Figure 2c with Figure R4c and included the information about statistical tests in figure legends (please see yellow highlighted parts on page 35 in the revised manuscript). Besides, the previous discussion “More importantly, D3.3 adsorbs more

A β 42 molecules on its surface and shows larger binding constant (K) than L3.3 ($1.8 \pm 0.9 \times 10^5$ vs. $1.4 \pm 0.5 \times 10^5$). The difference in binding free energy (ΔG) is ~ 0.2 kcal mol $^{-1}$ between D3.3 and L3.3.” have been rewritten as “The binding constant K between A β 42 and D3.3 is about 2.5 times larger than that between β 42 and L3.3. Furthermore, the Gibbs free energy change ΔG of binding process of A β 42 monomer on D3.3 is significantly lower than that on L3.3 (-7.3 ± 0.2 vs. -6.7 ± 0.2 kcal mol $^{-1}$, * $P < 0.05$), indicating the binding process of A β 42 with D3.3 is more energetically favorable.” (please see yellow highlighted parts on page 8 in the revised manuscript).

Comment 7. In cell experiments, it is unclear what aggregation state of amyloid-beta is used in cell treatments. The authors state in the methods that monomers are added to the cells; however, the protocol defined in the methods section for preparation of amyloid-beta (dilution of DMSO-dissolved amyloid-beta into aqueous buffer) is similar to that used for oligomer formation. Also, this preparation is added simultaneously with NPs; if the added preparation is oligomeric the role of NPs in preventing aggregation becomes less clear.

Response: Thanks a lot for the reviewer’s careful reading and insightful comment. To address the reviewer’s concern, the A β 42 solution that freshly prepared through our recipe has been characterized by size exclusion chromatography (SEC) (Figure R5a), dot blot assay (Figure R5b), and transmittance electronic morphology (Figure R5c). As a contrast, the other A β 42 sample that was generated after three hours of incubation at 37°C has been analyzed as well. As seen in figure R5a, the molecular weight of A β 42 form in the freshly prepared solution is about 8 kDa, which corresponds to that of A β 42 monomer reported before (*Proc. Natl. Acad. Sci.* **2010**, *107*, 15595-15600). After the A β 42 solution incubated at 37°C for 3 h, the molecular weight of A β 42 form increases to 33 kDa, indicating the formation of A β 42 oligomers (*Proc. Natl. Acad. Sci.* **2010**, *107*, 15595-15600) (Figure 5Ra). To better identify the nature of A β 42 forms in these two conditions, dot-blot analysis with A11 antibodies has been conducted. A11 is an antibody reported to selectively recognize soluble amyloid oligomers and prefibrillar aggregates (*Science* **2012**, *335*, 1228-1231). As seen in Figure R5b, the freshly prepared A β 42 solution produces a very little signal. In contrast, the A β 42 sample generated after three hours of incubation at 37°C shows strong positive signals, implying that the A β 42 solution prepared by our protocol remains

monomer instead of forming oligomers. This conclusion can be further verified by TEM images (Figure 5Rc). No large aggregates are observed in freshly prepared A β 42 solution. Meanwhile, A β 42 incubated at 37°C for 3 h generates sphere-shaped aggregates.

Figure R5. Validation of A β 42 preparation. (a) Size exclusion chromatography (SEC), (b) dot-blot assay, and (c) TEM images of freshly prepared A β 42 solution and A β 42 solution incubated at 37°C for 3 h.

Revision: We have added Figure R5 into the revised Supplementary Information as Supplementary Figure 6 (please see the yellow highlighted part on page S12). Besides, the experimental protocols for SEC and dot-blot assay are displayed on the yellow highlighted parts on page 18 and 19 in the revised manuscript.

Comment 8. On page 11, several differences in the biodistribution of D3.3 and L3.3 are discussed. It would be useful if these differences were supported by statistical indications within Figure 4.

Response: Thanks a lot for the reviewer's careful reading and insightful suggestion. Following the

reviewer's constructive suggestion, we have conducted statistical analyses for Figure 4. The results are presented in Figure R6. One can see that the biodistribution of D3.3 and L3.3 in the brain at first 6 h differs significantly, better verifying that the surface chirality of Au NPs strongly influences their distribution in the brain.

Figure R6. Biodistribution and toxicity evaluation of L3.3 and D3.3 *in vivo*. a,b, Biodistribution of L3.3 and D3.3 in the brain (a) and blood (b) at 6 h, 12 h, 24 h, and 48 h post-injection. Error bars indicate the s.d. (n = 4 mice per group). *P < 0.05; NS, not significant; Student's t-test. c, Elimination kinetics of L3.3 and D3.3 in plasma. The concentration-time profiles of L3.3 and D3.3 in plasma (b) fit a first-order elimination. The fitting equation for L3.3 and D3.3 groups is $y = 1.396 - 0.0211x$ ($r^2 = 0.992$) and $y = 1.43758 - 0.0203x$ ($r^2 = 0.989$), respectively. d, Hematological parameters, including red blood cells (RBC), hemoglobin (HGB), platelets (PLT) and white blood cells (WBC) of the KM mice after treatment with L3.3 or D3.3 for 6 h, 12 h, 24 h, and 48 h. The shadow regions represent the normal range of control groups. Error bars indicate the s.d. (n = 3 mice per group). e, H&E staining assays of the heart, liver, spleen,

lung, and kidney from all experimental groups (n = 2 mice per group).

Revision: We have replaced the original Figure 4 with Figure R6 and included the information about statistical tests in the figure legends of revised manuscript. (please see the yellow highlighted parts on page 37).

Comment 9. On page 12, some “significant” differences in behavior for mice treated with D3.3 and L3.3 are discussed (latency, time within target quadrant, platform crossings). Again, it would be useful if these significant differences were indicated within the figures.

Response: Thanks a lot for the reviewer’s careful reading and insightful suggestion. Following the reviewer’s constructive suggestion, we have added statistical analyses for the behavior of D3.3- and L3.3-treated AD mice. As shown in Figure R7, the latency of D3.3-treated AD mice is significantly lower than that of L3.3-treated AD mice (Figure R7a), the time that D3.3-treated AD mice spending within target quadrant is slightly longer than L3.3-treated AD mice (Figure R7b), and the platform crossing time of D3.3-treated AD mice is significantly more than L3.3-treated AD mice (Figure R7c). More importantly, the animals treated with D3.3 display remarkable improvement in spatial learning and memory in comparison with AD model mice, while L3.3 treatment does not cause significant improvement. Altogether, these data verify the better performance of D3.3 on rescuing memory deficits in AD model mice in comparison with L3.3.

Figure R7. Effect of L3.3 and D3.3 on rescuing memory impairments in AD model mice. a, Latency for escape to platform in the training phase. Error bars indicate the s.d. (n = 6 mice per group). **P < 0.01; ***P < 0.001; NS, not significant; one-way ANOVA. **b,** Percentage (%) of time spent in the target quadrant in probe test. Error bars indicate the s.d. (n = 6 mice per group). **P < 0.01, one-way ANOVA. **c,** Number of crossing platform time in probe test. Error bars indicate the s.d. (n = 6 mice per group). *P < 0.05; **P < 0.01; ***P < 0.001; NS, not significant;

one-way ANOVA.

Revision: We have replaced the original Figure 5a-c with Figure R7a-c and included the information about statistical tests in the figure legends of the revised manuscript. (please see the yellow highlighted parts on page 38). Further, to better describe the statistical results, we have added several sentences in the discussion. The renewed descriptions are as below: “During the five-day training and subsequent probe trial, the animals treated with D3.3 display remarkable improvement in spatial learning and memory of in comparison with AD model mice, while L3.3 treatment does not cause significant improvement (Fig. 5a-d and Supplementary video 1-8). Specifically, the animals administrated with D3.3 exhibit significantly shorter latency to escape onto the platform, slightly longer swimming time at the platform location, more platform crossing and better-focused search strategies for the platform in the quadrant compared with those treated with L3.3 (Fig. 5a-d)”. (please see the yellow highlighted parts on page 12 and 13).

Comment 10. Within the Conclusions, the authors discuss the superiority of their NPs over other NPs described in the literature. A table within the supplementary information is included to support this discussion. However, the discussion within the manuscript itself is limited. This argument would be much stronger if a more thorough discussion were included within the manuscript itself.

Response: Thanks a lot for the reviewer’s careful reading and insightful suggestion. Following the reviewer’s constructive suggestion, we have added several sentences in the revised manuscript to make the discussion about the superiority of our NPs becoming much stronger.

Revision: Several sentences have been added to the conclusion part. “In detail, our chiral NPs can cross the BBB through simple intravenous injection. We notice that the currently reported NP systems are either through stereotactic brain injection or intravenous injection with the assistance of cyclosporine A to modulate P-glycoprotein function. The operation of stereotactic brain injection is complicated and the mortality rate is high. Besides, modulating P-glycoprotein function to improve cerebral uptake has severe adverse effects that include an increased risk for serious infections. The last but not the least, the target quadrant occupancy of D3.3-treated AD

mice is closest to that of WT group among all the reported NPs (up to 94.9%).” (please see the yellow highlighted parts on page 14 and 15 in the revised manuscript).

Comment 11. Throughout the manuscript, a number of grammatical errors should be corrected.

Response: Thanks a lot for the reviewer’s careful reading and kind reminding. We have carefully checked the whole manuscript and supplementary information and revised many grammatical errors. We hope that the current version is easy to read.

Reviewer #3 (Remarks to the Author):

General Comments: The manuscript by Hou and colleagues describe the synthesis, characterization and utility of GSH-AU nanoparticles for potential treatment of Alzheimer's disease. Overall, the manuscript lacks adequate statistical analyses for the data presented and thus over-interpretation of the data is presented in both the results and discussion sections.

Response: We are very thankful for the reviewer's careful reading and critical comment, which help us to greatly improve the quality of this manuscript. We totally agree with the reviewer's comment that our statistical analyses for the data presented in the previous version are not adequate, which cause over-interpretation of the data to some extent. We have revised our manuscript according to the reviewer's insightful suggestions as below.

Comment 1. Figure 1. The secondary structure calculations in panel C do not match the representative CD spectra presented in panel b. This needs to be reconciled. There appears to be an over-estimation of the turn and random structures in both L3.3 and D3.3 + Ab42 results. Finally, the line estimation for Ab42+/-C3.5 does not accurately reflect the data or images in panels d and e.

Response: Thanks a lot for the reviewer's careful reading and insightful suggestion. Following the reviewer's kind comment, we have carefully checked the calculation methods for secondary structure and reconciled the helix, beta, turn and random structures in A β 42 and various A β 42 + Au NP systems. The results are summarized in below Figure R8. The A β 42 fibrils contain $48.3 \pm 3.0\%$ of β -sheet structure, which reduces to $45.2 \pm 3.4\%$, $43.2 \pm 0.4\%$, $29.3 \pm 2.1\%$, $47.4 \pm 2.2\%$, $37.3 \pm 0.5\%$, $47.4 \pm 0.4\%$, and $42.2 \pm 2.0\%$ upon incubation with C3.5, L3.3, D3.3, L9, D9, L15 and D15, respectively. Correspondingly, the percentage of the random coil structures increases. The A β 42 fibrils contain $17.6 \pm 1.4\%$ of β -sheet structure, which increases to $20.3 \pm 0.5\%$, $30.0 \pm 0.3\%$, $38.8 \pm 1.7\%$, $27.4 \pm 1.2\%$, $33.4 \pm 1.1\%$, $27.3 \pm 0.4\%$, and $29.9 \pm 1.0\%$ upon incubation with C3.5, L3.3, D3.3, L9, D9, L15 and D15, respectively. Furthermore, an indication of significance within the panels is included in Figure R8.

As for line estimation for A β 42+/-C3.5 in panels d and e of Figure 1, we would use it to discern morphology change of A β 42 aggregates (*Angew. Chem. Int. Ed.* **2011**, *50*, 5110-5115). Both AFM

(panel d) and TEM (panel e) images show that the density per surface area of long A β 42 fibrils slightly decreases with little increase of amorphous aggregates upon addition of C3.5. We also notice that such change is not so obvious compared with the samples of A β 42+L3.3 and A β 42+D3.3. We would like politely point out that all these observations (panels d and e) are consistent with fluorescence assay (panel a) and CD analysis (panels b and c).

Figure R8. **a**, Analysis of protein secondary structure of A β 42 in the absence and presence of L3.3, D3.3, or C3.5 after co-incubation for 48 hours. **b**, Analysis of protein secondary structure of A β 42 in absence and presence of L9, D9, L15 or D15 after co-incubation for 48 hours. Error bars indicate s.d. (n = 3 independent samples). *P < 0.05, **P < 0.01, ***P < 0.001, Student's t-test.

Revision: We have replaced original Figure 1c and Supplementary Figure 7c with Figure R8a and Figure R8b, respectively. In addition, the corresponding discussion in the revised manuscript has been updated as below. "The A β 42 fibrils contain $48.3 \pm 3.0\%$ of β -sheet structure, which reduces to $45.2 \pm 3.4\%$, $43.2 \pm 0.4\%$, $29.3 \pm 2.1\%$, $47.4 \pm 2.2\%$, $37.3 \pm 0.5\%$, $47.4 \pm 0.4\%$, and $42.2 \pm 2.0\%$ upon incubation with C3.5, L3.3, D3.3, L9, D9, L15 and D15, respectively. (please see the yellow highlighted parts on page 7 in the revised manuscript).

Comment 2. No evidence is given for the statement that "fluorescence quenching of ThT caused by AU-NPs can be ignored."

Response: Thanks a lot for the reviewer's kind reminding and we are sorry for our carelessness. Following the reviewer's suggestion, we have measured the fluorescence spectra of ThT aqueous solution (2 μ M) in the absence and presence of L3.3 (110 nM) or D3.3 (110 nM). As shown in Figure R9, the added L3.3 or D3.3 does not cause an obvious decrease on the fluorescence of ThT

solution. Thus, we conclude that the quenching raised by gold nanoparticles under the experimental conditions can be ignored. What's more, we have fully investigated the literatures, which support that the quenching effect of gold-based nanosystems on the fluorescence of ThT can be ignored. (*Small* **2012**, *8*, 3631-3639; *Chem. Eur. J.* **2015**, *21*, 829-835; *Theranostics* **2017**, *7*, 2996-3006.)

Figure R9. Fluorescence spectra of ThT aqueous solution (2 μM) in the absence and presence of L3.3 (110 nM) or D3.3 (110 nM). The excitation wavelength was 430 nm.

Revision: We have added Figure R9 into the revised Supplementary Information as Supplementary Figure 9, and also cited the literature (*Small* **2012**, *8*, 3631-3639) as “Considering the fact that the fluorescence quenching of ThT caused by Au NPs can be ignored under the experimental condition¹² (Supplementary Fig. 9)” in the revised manuscript. (please see the yellow highlighted part on the middle of page 6).

Comment 3. Figure 2a- The binding constants and delta G are not different statistically between L3.3 and D3.3, however the discussion in the results state they are. This needs fixed as does all subsequent discussion and implications of this data to the toxicity results.

Response: Thanks a lot for the reviewer's careful reading and insightful suggestion. We are sorry for our carelessness. The error bars that we gave in the previous manuscript represented the fitting errors in one experiment. To address the review's concern, we have repeated the ITC experiment for titration of A β 42 into the solution of L3.3 (Figure R10a) and D3.3 (Figure R10b) for three

times to ensure correctness of the result. The mean values and error bars for binding stoichiometry n_0 , binding constant K , enthalpy changes ΔH , entropy changes ΔS and Gibbs free energy changes ΔG are summarized in Figure R10c. The statistical analyses are also conducted. The results indicate that the binding constant K between A β 42 and D3.3 is about 2.5 times larger than that between β 42 and L3.3. Furthermore, the Gibbs free energy change ΔG of binding process of A β 42 monomer on L3.3 and D3.3 exhibits a significant difference (* $P < 0.05$), which indicates the binding process of A β 42 with D3.3 are more energetically favorable.

Figure R10. Analysis of interaction between L3.3/D3.3 and A β 42 monomer. a,b, ITC data for titration of A β 42 (0.138 mM) into solution of L3.3 (1.5 μM) (a) and D3.3 (1.5 μM) (b) at 25°C. (Upper) raw data and (lower) integrated data. The solid curves represent the best-fit results using a one-site binding model. The experiment was repeated for three times. c, Summary of binding

stoichiometry n_0 , binding constant K , enthalpy changes ΔH , entropy changes ΔS and Gibbs free energy changes ΔG of binding process of A β 42 monomer on L3.3 (**a**) and D3.3 (**b**). Error bars indicate mean \pm standard deviation (s.d.). * indicates $P < 0.05$, Student's t-test.

Revision: We have replaced the original Figure 2c with Figure R10c and included the information about statistical tests in figure legends. (please see yellow highlighted parts on page 35 in the revised manuscript). Besides, the previous discussion “More importantly, D3.3 adsorbs more A β 42 molecules on its surface and shows larger binding constant (K) than L3.3 ($1.8 \pm 0.9 \times 10^5$ vs. $1.4 \pm 0.5 \times 10^5$). The difference in binding free energy (ΔG) is ~ 0.2 kcal mol $^{-1}$ between D3.3 and L3.3.” have been rewritten as “The binding constant K between A β 42 and D3.3 is about 2.5 times larger than that between β 42 and L3.3. Furthermore, Gibbs free energy change ΔG of binding process of A β 42 monomer on D3.3 is significantly lower than that on L3.3 (-7.3 ± 0.2 vs. -6.7 ± 0.2 kcal mol $^{-1}$, * $P < 0.05$), indicating the binding process of A β 42 with D3.3 is more energetically favorable.” (please see yellow highlighted parts on page 8 in the revised manuscript).

Comment 4. Figure 3c/d- no error bars or statistics were performed for these analyses- was this an $n=1$ experiment, if so it needs to be repeated and statistics performed.

Response: Thanks a lot for the reviewer's careful reading and critical comment. Following the reviewer's constrictive suggestion, the flow cytometry analyses have been conducted for three independent times and the raw data are shown in Figure R11a. The population of apoptosis cells is summarized in Figure R11b and then the statistical analyses have been conducted. Compared with pure A β 42, incubation of A β 42 with L3.3 or D3.3 gives rise to a decrease of apoptosis from 42.9% to 36.2% or 32.1%, respectively. Both L3.3 and D3.3 can protect SH-SY5Y neuronal cells from cytotoxicity induced by A β 42 aggregation, and D3.3 has a more significant effect in reducing the apoptosis rate.

Figure R11. **a**, Raw data for flow cytogram assay based on annexin V-fluorescein isothiocyanate (Annexin V-FITC)/propidium iodide (PI) staining of SH-SY5Y cells after treatment with different therapeutic groups for 48 h. **b**, Population of apoptosis cells summarized from flow cytometry analysis. Error bars indicate the standard deviation (s.d.) (n = 3 independent experiments). *P < 0.05, **P < 0.01, ***P < 0.001, one-way ANOVA.

Revision: In the revised manuscript, we have replaced Figure 3c with Figure R11b and the corresponding discussion on page 10 has been updated as “Compared with pure Aβ42, incubation of Aβ42 with L3.3 or D3.3 gives rise to a decrease of apoptosis from 42.9% to 36.2% or 32.1%”. The

information about statistical tests have been included in the figure legends of the revised manuscript as well. (please see the yellow highlighted parts on the page 36).

Comment 5. Figure 4 there are no relevant differences between Au in brain and blood between L3.3 and D3.3 yet the results sections states differences. Neither is relevant and should be removed.

Response: Thanks a lot for the reviewer's careful reading and insightful comment. Following the reviewer's suggestion, we have performed statistical analyses for the distribution of L3.3 and D3.3 in the brain and blood. As shown in Figure R12a, the biodistribution of D3.3 and L3.3 in the brain at first 6 h differs significantly, indicating that the surface chirality of Au NPs strongly influences their distribution in the brain at the very beginning. However, this significant difference disappears with time prolonging. This may be ascribed to the fact that L3.3 and D3.3 begin to be cleared from the brain. It is also noted that the distribution of L3.3 and D3.3 in the blood does not present a significant difference (Figure R12b), although the amount of D3.3 looks slightly higher than that of L3.3. Following the review's suggestion, we have deleted the discussion related to this part and rewritten the discussion.

Figure R12. Biodistribution and toxicity evaluation of L3.3 and D3.3 *in vivo*. a,b, Biodistribution of L3.3 and D3.3 in the brain (a) and blood (b) at 6 h, 12 h, 24 h, and 48 h post-injection. Error bars indicate the s.d. (n = 4 mice per group). *P < 0.05; NS, not significant; Student's t-test.

Revision: We have updated Figure 4a,b using Figure R12 and included the information about

statistical tests in the figure legends of revised manuscript. (please see the yellow highlighted parts on page 37). Also, the discussion related to Figure 4a,b has been rewritten as below: “As summarized in Fig. 4a, the amount of Au in the brain of L3.3- or D3.3-injected mice reaches the maximum concentration (10 times higher than that of the control group) at 12 h or 6 h, respectively. This result discloses that L3.3 and D3.3 could be transported from the blood circulation into the brain. Notably, the surface chirality of Au NPs significantly influences the biodistribution of D3.3 and L3.3 in the brain at first 6 h, implying that L3.3 and D3.3 may cross the BBB through transporter proteins⁴⁴ besides spontaneous penetration^{19,45}. Besides, the Au content begins to decrease after 12 h for L3.3-injected mice and 24 h for D3.3-injected mice, respectively, suggesting that the clearance of D3.3 from the brain is slower than that of L3.3. As expected, the amount of L3.3 and D3.3 in the blood gradually decreases with the prolonged time because of clearance and aggregation (Fig. 4b).” (please see the yellow highlighted part on page 11 in the revised manuscript).

Comment 6. Figure 5. Limited statistics were performed, data is over-interpreted. No differences in the L3.3 group from untreated mice and yet presented as differences. Should be fixed.

Response: Thanks a lot for the reviewer’s careful reading and critical comment, which greatly help us improve the quality of our manuscript. Following the reviewer’s kind suggestions, we have improved the statistical analyses for Figure 5. The results are shown in Figure R13. As seen from Figure R13a-c, the treatment of L3.3 does not give rise to significant improvement on the behavior of AD mice. However, L3.3 treatment leads to a significant decrease in the number of insoluble A β 42 and soluble A β 40 (Figure R13g,h). This means the treatment of L3.3 can decrease A β deposition, but the effect is not intense enough to lead to the obvious improvement in the behavior of AD mice.

Figure R13. Effect of L3.3 and D3.3 on rescuing memory impairments in AD model mice. a, Latency for escape to platform in the training phase. Error bars indicate the s.d. (n = 6 mice per group). **P < 0.01; ***P < 0.001; NS, not significant; one-way ANOVA. **b,** Percentage (%) of time spent in the target quadrant in probe test. Error bars indicate the s.d. (n = 6 mice per group). **P < 0.01; NS, not significant; one-way ANOVA. **c,** Number of crossing platform time in probe test. Error bars indicate the s.d. (n = 6 mice per group). *P < 0.05; ***P < 0.001; NS, not significant; one-way ANOVA. **d,** Track of probe test. **e,** Immunofluorescence for A β in the hippocampus of WT control mice, AD control mice, AD mice with L3.3 or D3.3 treatment. **f-i,** Soluble A β 42 (**f**), insoluble A β 42 (**g**), soluble A β 40 (**h**), and insoluble A β 40 (**i**) in the brains measured by ELISA. Error bars indicate the s.d. (n = 3 mice per group). *P < 0.05; **P < 0.01; ***P < 0.001; NS, not significant; one-way ANOVA.

Revision: We have renewed Figure 5 with Figure R13 and included information about statistical tests in figure legends of the revised manuscript. (please see the yellow highlighted parts on page 38). In the corresponding discussion part, we have rewritten that “During the five-day training and subsequent probe trial, the animals treated with D3.3 display remarkable improvement in spatial

learning and memory in comparison with AD model mice, while L3.3 treatment does not cause significant improvement (Fig. 5a-d and Supplementary video 1-8).” (please see the yellow highlighted parts on page 12). Also, we have added a sentence to make it clear that “Notably, the L3.3 treatment leads to a significant difference in the number of insoluble A β 42 and soluble A β 40, but this decrease is not intense enough to cause obvious improvement on the behavior of AD mice.” (please see the yellow highlighted parts on page 13).

Comment 7. For the discussion, since all Abeta-therapies have failed in the clinic some discussion on comparison of previous therapies to this should be presented at the preclinical level- to justify the statement that this is a better treatment option.

Response: Thanks a lot for the reviewer’s careful reading and insightful suggestion. The A β -based drugs suffer failures to date, likely in part because the drugs tested lack sufficient target engagement or have toxic effects (*Curr. Med. Res. Opin.* **2014**, *10*, 2021-2032). Our D-GSH stabilized Au NPs show much-improved prevention of A β aggregation via adsorption of peptide monomers on their curvature surfaces. Besides, the small size and GSH coating enable D3.3 crossing the blood brain barrier and good biosafety. All these merits imply that our D3.3 would be a better treatment option on the treatment of Alzheimer’s disease over previous therapies.

Revision: We have rewritten part of the discussion as “The A β -based drugs suffer failures to date, likely in part because the drugs tested lack sufficient target engagement or have toxic effects. We highly expect that chiral nanomedicine will open a new era toward the treatment of Alzheimer’s disease thanks to their size- and stereo-determined inhibition of peptide or protein aggregation, the capability of crossing the blood-brain barrier, good biosafety and rather rapid clearance from the body.”. (please see the yellow highlighted parts on page 15).

REVIEWERS' COMMENTS:

Reviewer #3 (Remarks to the Author):

The manuscript is much improved by inclusion of statistical analyses, however, it was not clear which post-hoc tests were utilized in the analyses presented. Furthermore, the Morris Water Maze must be analyzed using a repeated measures ANOVA with Holm-Sidak post hoc test to correct for multiple comparisons not a simple one way ANOVA. These data need to be analysed.

REVIEWERS' COMMENTS:

Reviewer #3 (Remarks to the Author):

The manuscript is much improved by inclusion of statistical analyses, however, it was not clear which post-hoc tests were utilized in the analyses presented. Furthermore, the Morris Water Maze must be analyzed using a repeated measures ANOVA with Holm-Sidak post hoc test to correct for multiple comparisons not a simple one way ANOVA. These data need to be analysed.

Response: Thanks for the reviewer's insightful suggestions. We are sorry for not clarifying in our previous manuscript, and the one-way ANOVA with the least significant difference (LSD) post hoc test was utilized. Following the reviewer's suggestions, we re-analyzed the data presented in the Figure 3 and Figure 5 (Morris Water Maze), using the one-way ANOVA analyses with Holm-Sidak post hoc test. The results has been updated in the Figure 3 and 5. The statistic method has been described in the relevant Figure legends and Method section. The detailed P values have been also updated in Supplementary Table 7. It should be noted that the new statistic method does not change the discussion and conclusion in our manuscript.